# Efficient Utility-Preserving Machine Unlearning with Implicit Gradient Surgery

**Shiji Zhou[1,2]*†, Tianbai Yu[3], Zhi Zhang[4], Heng Chang[5], Xiao Zhou[5], Dong Wu[6], Han Zhao[3]**

[1]Institute of Artificial Intelligence, Beihang University
[2] Beijing Advanced Innovation Center for Future Blockchain and Privacy Computing, Beihang University
[3]University of Illinois at Urbana-Champaign [4]University of Amsterdam
[5]Tsinghua University [6]YanTron Technology Co.Ltd

## Abstract

Machine unlearning (MU) aims to efficiently remove sensitive or harmful memory from a pre-trained model. The key challenge is to balance the potential tradeoff between unlearning efficacy and utility preservation, which involves forgetting undesirable information as defined while maintaining the model's original performance. One potential way to tackle this problem is to use multi-objective optimization to jointly optimize both the unlearning and utility preservation objectives. However, existing multi-objective methods only guarantee finding a Pareto-optimal solution without fine-grained control, which causes under-optimization of the unlearning objective. To this end, we first model MU as a constrained optimization problem, that is, optimizing the unlearning objective under the constraint of a bounded increase for utility loss. We then show that solving this optimization problem is equivalent to unilateral gradient surgery on the unlearning objective. To resolve the additional computational cost brought by gradient surgery, we propose an implicit gradient surgery method, which approximates the solution to the aforementioned constrained optimization problem via only one backpropagation, thereby achieving efficient utility-preserving MU. Theoretically, we provide a tight convergence analysis of the algorithm. Empirically, our extensive experiments show that the proposed algorithm achieves better tradeoff results than existing baselines. Codes are available at `https://github.com/anseryuer/EUPMU-Efficient-Utility-Preserving-Machine-Unlearning`.

## 1 Introduction

The growing capacity of large generative models [39, 5, 74] has inevitably led to increasing concerns about their potential security risks. In particular, massive pre-training data from large models may contain privacy, copyright, and illegal information about individual users, which can be inadvertently memorized through model parameters through training, posing a risk of content leakage under model inversion attacks [21, 71]. Moreover, the high training costs of large models make addressing these issues in pre-trained models particularly challenging [52], since naive retraining is computationally infeasible. Consequently, both industry and academia are actively seeking efficient methods to enable the erasing of sensitive information at a small cost.

Machine unlearning (MU) [4], which aims to enable models to efficiently remove memory of sensitive data, is a potential approach to meet the above goals. The current landscape of MU

---

*Part of the work was done while Shiji Zhou was at Tsinghua University
†Corresponding to Shiji Zhou: zhoushiji25@buaa.edu.cn
Shiji Zhou and Tianbai Yu have equal contributions

39th Conference on Neural Information Processing Systems (NeurIPS 2025).

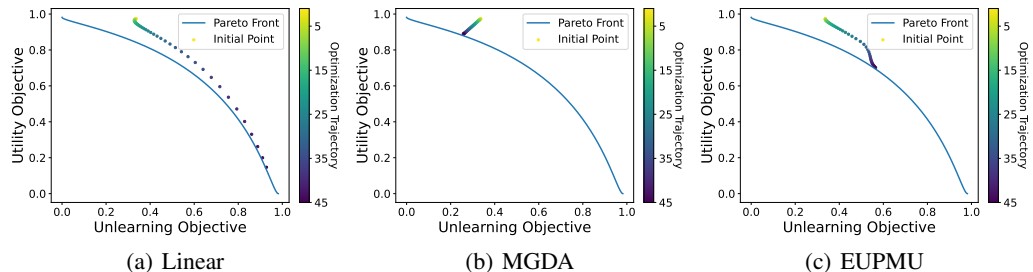

| (a) Linear | (b) MGDA | (c) EUPMU |

Figure 1: A showcase where both linear scalarization and MGDA fail to find a good tradeoff solution that balance the utility and unlearning objectives: (a) Linearization often leads to extreme solutions with deterioration on one objective; (b) The pre-trained model fully optimizes the utility objective, leaving little room to improve unlearning objective for MGDA algorithm that fairly optimizes all objectives; (c) Our EUPMU algorithm provides a tolerance for small degradation for utility objective, and returns a better tradeoff solution.

research encompasses a range of generative models, such as large language models (LLM)[68], image synthesis models [50], and multi-modal generative frameworks [59]. These investigations have showcased the capacity of MU to eliminate specific data, including copyrighted patterns [22], fake personas [17], and confidential information [60]. However, the utility-unlearning tradeoff is a key issue in MU [47], where there is a fundamental tradeoff between enhancing the unlearning effect and maintaining the model's original performance. This often leads to a degradation of the model's performance when unlearning sensitive information. Current solutions involve incorporating a retaining loss, calculated from the retained portion of the training data, into the unlearning training process, which can alleviate some of the issues with utility degradation [1, 18]. However, simply combining unlearning and retaining objectives that have inherent conflict fails to find a balanced solution, as it has been proved in multitask learning literature [34], as demonstrated in Figure 1(a).

A potential way to mitigate the conflicts between the two targets is to employ a multi-objective optimization (MOO) [77]. However, existing MOO methods cannot control the converged solutions in a fine-grained way. Specifically, pre-trained models have already been thoroughly optimized for utility, hence it is already or near to a Pareto optimal solution on the Pareto front between unlearning and utility. Since there is little room for further optimization of the utility objectives, directly applying MOO methods that often fairly optimize all objectives to the unlearning problem may result in insufficient optimization of the unlearning objectives, as demonstrated in Figure 1(b). Moreover, most existing MOO methods require derivatives for each objective, doubling the number of backpropagation iterations compared to linear weighting methods, thereby doubling the computational cost. This contradicts the high efficiency and low-cost demand of unlearning.

To tackle the challenges, this paper makes the following principal contributions:

1) To better address the utility-unlearning tradeoff, we first formulate the *Utility-Preserving Unlearning Problem (UPUP)*, where the preservation of utility is formulated as a constraint on the increase in retaining loss at each step, with the goal of maximizing the decrease in unlearning loss under this constraint, thereby optimizing the unlearning objective while preserving utility. Solving UPUP leads to a gradient method equivalent to *Unilateral Gradient Surgery*, which subtracts the components of the unlearning gradient that conflict with the retaining gradient. This ensures that utility degradation is within a controllable range while maxing the erasing of target information.

2) Explicit gradient surgery inevitably doubles the gradient computation. This paper first presents an *Efficient Utility-Preserving Machine Unlearning (EUPMU)* method with *Efficient Implicit Gradient Surgery*, utilizing the essence of gradient surgery being equivalent to dynamic linear weighting. Solving the weights through a first-order approximation, requires only one pass of gradient computation, thus saving up to 50% of the main computational costs. Furthermore, we provide a multi-objective convergence analysis for the proposed algorithm, proving that the algorithm can efficiently converge to a Pareto optimal (or stationary) solution, demonstrating that the algorithm can achieve sufficient unlearning while preserving utility.

3) We conducted comprehensive experiments on tasks including image classification and image generation. Both numerical and visual results demonstrated significant improvements, effectively

proving that our method can fully optimize the unlearning objective while maintaining utility to the greatest extent. This further validates the design and theoretical outcomes of our approach.

## 2 Background

We begin by introducing the basic concepts of machine unlearning, elucidating the utility-unlearning challenge encountered by existing methods. We then present the concept of Multi-Objective Optimization (MOO) as a potential solution to this problem and discuss why MOO cannot be directly applied to address the challenges in MU. We defer detailed comparison with related works in Appendix A.

### 2.1 Machine Unlearning

Given the *training dataset* $\mathcal{D} = \{z_i = (x_i, y_i)\}_{i=1}^N$, the original/pre-trained model with parameters $\boldsymbol{\theta}_0$ is encapsulated by the following optimization:

$$\boldsymbol{\theta}_0 = \arg\min_{\theta} \mathbb{E}_{z \sim \mathcal{D}} \ell(z; \boldsymbol{\theta}).$$

MU focuses on eliminating the influence of a specific data subset $\mathcal{D}_f \subset \mathcal{D}$, which is the *forgetting dataset* that may include harmful or sensitive information. The goal is to efficiently derive an unlearned model $\theta_u$ by finetuning the original parameters $\boldsymbol{\theta}_0$, while maintaining the model's performance on the *retaining dataset* $\mathcal{D}_r = \mathcal{D} \setminus \mathcal{D}_f$.

To counteract the impact of the forgetting dataset $\mathcal{D}_f$, MU algorithms often establish objectives that aim to reverse the effects of the initial training[3]. These include objectives such as inverse loss [61] and random labeling [24]. We denote this corrective objective as the *unlearning objective* $\ell_u(\boldsymbol{\theta})$. Furthermore, to maintain performance on the remaining data, most MU methods incorporate an objective that mimics retraining on samples from the retaining dataset $\mathcal{D}_r$. This is known as the *retaining objective* $\ell_r(\boldsymbol{\theta})$. Typically, these objectives are linearly combined into one objective [18]:

$$\ell_u(\boldsymbol{\theta}) + \lambda \ell_r(\boldsymbol{\theta}),$$

where $\lambda$ is a hyperparameter balancing the two parts.

**Utility-Unlearning Challenge.** While linearization is straightforward to implement, it may lead to performance deterioration or insufficient unlearning, due to the inherent conflict between the unlearning and retaining objectives [72, 75]. The underlying theoretical reason may be that fixed linear weights fails to find a solution that balances the two objectives [34], as shown in Figure 1. (a).

### 2.2 Multi-Objective Optimization

Multiple-objective optimization (MOO) aims to optimize multiple objectives simultaneously [77]:

$$\min_{\boldsymbol{\theta}} \boldsymbol{L}(\boldsymbol{\theta}) = (\ell^1(\boldsymbol{\theta}), \dots, \ell^m(\boldsymbol{\theta}))^\top, \tag{1}$$

where $m \geq 2$ denotes the number of objectives, and $\ell^i : \mathbb{R}^n \to \mathbb{R}$ is the $i$-th loss function. Denote $\Delta_m$ to be the $(m-1)$-dimensional probability simplex. The concept of Pareto optimality/stationary [12] is introduced to determine whether a solution to MOO is optimal/critical.

**Definition 2.1. Pareto Optimality:** For any two solutions $\boldsymbol{\theta}, \boldsymbol{\theta}'$, we say that $\boldsymbol{\theta}$ dominates $\boldsymbol{\theta}'$, denoted as $\boldsymbol{\theta} \prec \boldsymbol{\theta}'$ or $\boldsymbol{\theta}' \succ \boldsymbol{\theta}$, if $\ell^i(\boldsymbol{\theta}) \leq \ell^i(\boldsymbol{\theta}')$ for all $i$, and there exists one $i$ such that $\ell^i(\boldsymbol{\theta}) < \ell^i(\boldsymbol{\theta}')$. A solution $\boldsymbol{\theta}^* \in$ is called Pareto optimal if it is not dominated by any other solution. **Pareto Stationary:** A solution $\boldsymbol{\theta}$ is called Pareto stationary if there exists $\boldsymbol{\lambda} \in \Delta_m$ such that $\sum_{i=1}^m \lambda_i \nabla_{\boldsymbol{\theta}} \ell_i(\boldsymbol{\theta}) = 0$.

Typical MOO methods like MGDA [14], PCGrad [69] and other variants [46, 76, 45, 28] aim to search for a direction $\boldsymbol{d}$ that is not conflicting with each gradient, i.e., $\nabla \ell^i(\boldsymbol{\theta})^\top d \geq 0, i \in [m]$. Using such a non-conflicting direction $d_k$ as the update direction is shown to get better tradeoff performance.

**Can MOO Solve Utility-Unlearning Tradeoff?** MOO addresses conflicts among objectives during the optimization process and may offer a solution to the utility-unlearning challenge in MU. However,

---

[3]We here consider approximate unlearning, since exact unlearning often needs to retrain the model, which is impractical for large models due to the high retraining cost.

the optimization goals of MOO and MU are not aligned, making existing MOO methods inadequate for unlearning in pre-trained models. Specifically, pre-trained models have been thoroughly optimized for the utility objective, while the unlearning objective has not been optimized. Therefore, the goal of unlearning is to fully optimize the unlearning objective while maintaining utility to the greatest extent possible, albeit with some minor degradation if necessary, like the solution in Figure 1. (c). However, the primary goal of typical MOO methods is to identify an optimization path that benefits all objectives simultaneously. Given that there is little room for further improvement in utility objectives, directly applying MOO methods to optimize utility and unlearning objectives fairly could lead to inadequate optimization of the unlearning objective, as shown in Figure 1. (b). In addition, MOO often requires gradient computation for each objective, which doubles the computational cost of linearization, which only needs to compute the gradient of the linearized loss.

## 3   Efficient Utility-Preserving Machine Unlearning

This section first presents the formulation of the Utility-Preserving Unlearning Problem (UPUP). Then, we introduce a gradient method for solving UPUP, which is equivalent to explicit unilateral gradient surgery. To address the additional computational cost brought by gradient surgery, we propose a method of implicit efficient gradient surgery, an efficient approximation for solving UPUP (Algorithm 1). Finally, we provide a theoretical analysis of the Pareto optimality/stationary.

### 3.1   Utility-Preserving Unlearning Problem

At iteration $t$, we perform the update $\boldsymbol{\theta}_{t+1} = \boldsymbol{\theta}_t - \alpha_t \boldsymbol{d}_t$ where $\boldsymbol{d}_t$ is the update direction, and define the improvement of the retaining and unlearning objectives as follows

$$r_r(\alpha_t, \boldsymbol{d}_t) = \ell_r(\boldsymbol{\theta}_t) - \ell_r(\boldsymbol{\theta}_{t+1}), r_u(\alpha_t, \boldsymbol{d}_t) = \ell_u(\boldsymbol{\theta}_t) - \ell_u(\boldsymbol{\theta}_{t+1}).$$

To achieve utility-preserving unlearning, we aim to seek a direction $\boldsymbol{d}_t$ to control the degradation of the local retaining target $r_r$ during the constrained optimization process while maximizing the optimization of the unlearning objective. Mathematically, this can be expressed as:

$$
\begin{aligned}
\max_{\boldsymbol{d}_t} \quad & \frac{1}{\alpha_t} r_u(\alpha_t, \boldsymbol{d}_t) - \frac{1}{2} \|\boldsymbol{d}_t\|^2 \\
\text{s.t.} \quad & \frac{1}{\alpha_t} r_r(\alpha_t, \boldsymbol{d}_t) \geq -\varepsilon_t,
\end{aligned}
\tag{2}
$$

where $\alpha_t$ is the stepsize, and $\|\boldsymbol{d}_t\|^2$ is the regularization to avoid unbounded solutions. Here, $\varepsilon_t \geq 0$ is the tolerance of the degradation for the retaining target that we aim to preserve, and the constraint $\frac{1}{\alpha_t} \cdot r_r(\alpha_t, \boldsymbol{d}_t) \geq -\varepsilon_t$ ensures that the utility performance drop is controllable.

### 3.2   Explicit Unilateral Gradient Surgery

Since stepsize $\alpha_t$ is usually small, by the first-order Taylor approximation, we know that

$$r_r(\alpha_t, \boldsymbol{d}_t) \approx \alpha_t \nabla \ell_r(\boldsymbol{\theta}_t) \cdot \boldsymbol{d}_t, r_u(\alpha_t, \boldsymbol{d}_t) \approx \alpha_t \nabla \ell_u(\boldsymbol{\theta}_t) \cdot \boldsymbol{d}_t.$$

Problem 2 can be approximated by

$$
\begin{aligned}
\max_{\boldsymbol{d}_t} \quad & \nabla \ell_u(\boldsymbol{\theta}_t) \cdot \boldsymbol{d}_t - \frac{1}{2} \|\boldsymbol{d}_t\|^2 \\
\text{s.t.} \quad & \nabla \ell_r(\boldsymbol{\theta}_t) \cdot \boldsymbol{d}_t \geq -\varepsilon_t.
\end{aligned}
\tag{3}
$$

Problem 3 aims to control the dot product of $\boldsymbol{d}_t$ and $\nabla \ell_r$ to be greater than $-\varepsilon_t$, thereby ensuring that the degradation of $\ell_r$ is less than $\varepsilon_t$, while simultaneously maximizing the dot product of $\boldsymbol{d}_t$ and $\nabla \ell_u$ to achieve more effective unlearning.

**Proposition 3.1.** *The dual objective of Problem 3 is*

$$\min_{\lambda_t \geq 0} L_t(\lambda_t) = \frac{1}{2} \|\nabla \ell_u(\boldsymbol{\theta}_t) + \lambda_t \nabla \ell_r(\boldsymbol{\theta}_t)\|^2 + \lambda_t \varepsilon_t.
\tag{4}$$

Problem 4 has a closed form solution, and the desired direction $\boldsymbol{d}_t^*$ can be solved as

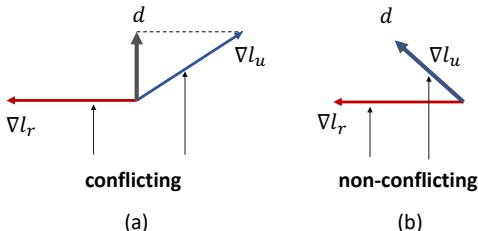

Figure 2: Illustration of Unilateral Gradient Surgery. The computation of the update direction is divided into two scenarios: (a) If the gradient of the unlearning objective conflicts with the gradient of the retaining objective, that is, the angle between them is greater than 90 degrees, the desired direction is obtained by removing the projection (to the retaining gradient) component from the unlearning gradient; (b) If the gradient of the unlearning objective does not conflict with the gradient of the retaining objective, then the unlearning gradient itself is the desired direction.

Figure 3: Illustration of the role of $\varepsilon_t$. (a) When $\varepsilon_t = 0$, if the retaining gradient and the unlearning gradient are opposite in direction, the update direction $d = 0$, at which point optimization halts and no further progress can be made in optimizing the unlearning objective. (b) When $\varepsilon_t \neq 0$, the update direction $d \neq 0$, and optimization of the unlearning objective proceeds under the condition that it does not unduly affect the retaining objective.

**Proposition 3.2.** *The closed form solution of Problem 3*

$$\boldsymbol{d}_t^* = \begin{cases} \nabla\ell_u(\boldsymbol{\theta}_t) + \lambda_t^* \nabla\ell_r(\boldsymbol{\theta}_t), & \textit{if } \lambda_t^* > 0 \\ \nabla\ell_u(\boldsymbol{\theta}_t), & \textit{if } \lambda_t^* \leq 0 \end{cases} \tag{5}$$

*where*

$$\lambda_t^* = \frac{-\nabla\ell_r(\boldsymbol{\theta}_t) \cdot \nabla\ell_u(\boldsymbol{\theta}_t) - \varepsilon_t}{\|\nabla\ell_r(\boldsymbol{\theta}_t)\|^2}. \tag{6}$$

Due to space limit, we defer the derivation details to the Appendix C.2 & C.3. It is worth noting that when $\varepsilon_t = 0$, the direction of $\boldsymbol{d}_t^*$ is equivalent to performing unilateral gradient surgery on the gradient of the unlearning objective, $\nabla\ell_u(\boldsymbol{\theta}_t)$. As illustrated in Figure 2, when $\nabla\ell_u(\boldsymbol{\theta}_t)$ is in conflict with the gradient of the retaining objective, $\nabla\ell_r(\boldsymbol{\theta}_t)$, the direction $\boldsymbol{d}_t^*$ is obtained by eliminating the conflicting component of $\nabla\ell_u(\boldsymbol{\theta}_t)$ along $\nabla\ell_r(\boldsymbol{\theta}_t)$. When $\nabla\ell_u(\boldsymbol{\theta}_t)$ and $\nabla\ell_r(\boldsymbol{\theta}_t)$ are not in conflict, then $\boldsymbol{d}_t^* = \nabla\ell_u(\boldsymbol{\theta}_t)$, and directly optimizing the unlearning objective will not have a negative impact on the retaining objective.

When $\varepsilon_t \neq 0$, an error tolerance is introduced, gradient surgery is only performed when the conflict between the unlearning and the retaining gradients is sufficiently large, that is, when $\nabla\ell_r(\boldsymbol{\theta}_t) \cdot \nabla\ell_u(\boldsymbol{\theta}_t) < -\varepsilon_t$. This threshold controls the priority during the optimization process for the unlearning objective. Specifically, when the angle between the retaining gradient and the unlearning gradient is 180 degrees as illustrated in Figure 3, if $\varepsilon_t = 0$, the optimization process stops and the unlearning objective can not be further optimized. In contrast, when $\varepsilon_t \neq 0$, a certain degree of optimization for the unlearning objective is still guaranteed.

*Remark* 3.3. We can demonstrate that by imposing local constraints on the retaining objective at each training step, the decline in utility throughout the entire optimization process can be controlled by

$$\ell_r(\boldsymbol{\theta}_t) - \ell_r(\boldsymbol{\theta}_0) \lesssim \mathcal{O}\left(\sum_{i=1}^{t} \varepsilon_t \alpha_t\right). \tag{7}$$

Proof details are deferred to the Appendix C.4. This illustrates that the increase in the retaining objective is controlled by $\sum_{i=1}^{t} \varepsilon_t \alpha_t$. This indicates that we can achieve the goal of utility-preservation by adjusting the hyperparameters.

## 3.3 Implicit Efficient Gradient Surgery

Although unilateral gradient surgery can ensure that the optimization results meet the expected utility-preserving criteria, the computational cost in the optimization process is double that of the linear weighting method due to the need to calculate gradients for both the unlearning and retaining objectives separately. This can lead to inefficiency in optimization, which contradicts the high-efficiency and low-cost goals that unlearning aims to achieve. To tackle this challenge, we propose an efficient approximate solution to Problem 3 in the following section, which requires only a single step of backpropagation, making the computational cost equivalent to that of the linear weighting method.

Since the solution to Problem 4 requires knowledge of the gradients for both the unlearning and retaining objectives, we consider using a gradient descent approximation to solve for $\lambda_t$:

$$\lambda_{t+1} = \lambda_t - \beta_t \nabla_{\lambda_t} L_t(\lambda_t).$$

However, the gradient of $L_t(\lambda_t)$ still necessitates information about the gradients of the unlearning and retaining objectives. Therefore, we consider approximating the gradient $\nabla_{\lambda_t} L_t(\lambda_t)$ using first-order Taylor's approximation:

$$\begin{aligned}
\nabla_{\lambda_t} L_t(\lambda_t) &= \nabla \ell_r(\boldsymbol{\theta}_t) \cdot (\nabla \ell_u(\boldsymbol{\theta}_t) + \lambda_t \nabla \ell_r(\boldsymbol{\theta}_t)) + \varepsilon_t \\
&= \nabla \ell_r(\boldsymbol{\theta}_t) \cdot \boldsymbol{d}_t + \varepsilon_t \\
&\approx \frac{1}{\alpha_t}(\ell_r(\boldsymbol{\theta}_t) - \ell_r(\boldsymbol{\theta}_{t+1})) + \varepsilon_t.
\end{aligned}$$

Consequently, we have an approximate method for solving $\lambda_t$ without backpropagation:

$$\begin{aligned}
&\lambda_{t+1} = \lambda_t - \beta_t \tilde{\delta}_t, \\
&\text{where } \tilde{\delta}_t = \frac{1}{\alpha_t}(\ell_r(\boldsymbol{\theta}_t) - \ell_r(\boldsymbol{\theta}_{t+1})) + \varepsilon_t.
\end{aligned} \tag{8}$$

We provide the complete Efficient Utility-Preserving Machine Unlearning (EUPMU) algorithm (Algorithm 1) in Appendix B. The algorithm first computes the approximated weight $\lambda_t$ by Eq.8 with no backpropagation of the loss function, and then uses $\lambda_t$ to compute the update direction $d_k$ by one backpropagation of the composite loss $\ell_u(\boldsymbol{\theta}_t) + \lambda_t \ell_r(\boldsymbol{\theta}_t)$. Finally, the model parameter is updated by one gradient step with $d_k$. This process does not require to compute $\nabla \ell_u(\boldsymbol{\theta}_t)$ and $\nabla \ell_r(\boldsymbol{\theta}_t)$ separately like traditional multi-objective method, and hence save half of the computational cost.

**A faster version of EUPMU.** We propose EUPMU-fast as a lightweight variant of EUPMU that removes the second retain-loss recomputation on the same batch. Concretely, whereas EUPMU estimates the retain-loss change at step $t$ as $\Delta_r^{(t)} = \ell_r(\theta_t; \mathcal{B}_r^{(t)}) - \ell_r(\theta_{t+1}; \mathcal{B}_r^{(t)})$, which requires an extra forward pass at $\theta_{t+1}$ on $\mathcal{B}_r^{(t)}$, EUPMU-fast uses a stochastic proxy based on consecutive retain batches:

$$\widehat{\Delta}_r^{(t)} \approx \ell_r(\theta_{t+1}; \mathcal{B}_r^{(t+1)}) - \ell_r(\theta_t; \mathcal{B}_r^{(t)}),$$

so it incurs no additional forward pass. This reduces wall-clock per step (see RTE) but can be less stable due to between-batch variation; in our runs, it sometimes underperforms EUPMU, though it remains competitive while being slightly faster.

## 3.4 Theoretical Analysis

Our primary concern is whether $\lambda_t$ in Algorithm 1 can approximate the property of the optimal solution $\lambda_t^*$, which determines whether efficient gradient surgery can achieve the goal of utility-preservation. Therefore, we present the following theorem to elucidate this result.

**Theorem 3.4** (Approximate $\lambda^*$). *Suppose retaining objective and unlearning objective are both (i) G-Smooth; (ii) L-Lipschitz. At training step t, setting $\sum_{i=0}^{t} \alpha_i \leq \mathcal{O}(1)$, $\beta_i/\alpha_i = \mathcal{O}(1/t^{1/3})$ and $\sum_{i=0}^{t} \varepsilon_i \leq \mathcal{O}(1)$, we have*

$$\frac{1}{t} \sum_{i=1}^{t} (L_i(\lambda_i) - L_i(\lambda_i^*)) \leq \mathcal{O}(1/t^{1/3}) \tag{9}$$

Table 1: Performance of class-wise forgetting on Imagenette using SD. The best performance is highlighted in **bold**.

| Forget.Class | SalUn | | ESD | | FMN | | EUPMU | |
|---|---|---|---|---|---|---|---|---|
| | UA | FID | UA | FID | UA | FID | UA | FID |
| Tench | 0.00 | 0.94 | 0.00 | 1.18 | 56.20 | **0.86** | 0.00 | 0.93 |
| EnglishSpringer | 0.00 | **0.79** | 0.00 | 0.98 | 71.40 | 1.24 | 0.00 | 1.52 |
| CassettePlayer | 0.20 | 1.59 | 3.40 | 1.75 | 11.20 | 1.02 | 0.00 | **0.97** |
| ChainSaw | 0.00 | 1.07 | 0.00 | 1.55 | 50.80 | **0.88** | 0.00 | 0.99 |
| Church | 0.40 | 0.99 | 2.60 | 1.88 | 75.60 | 1.66 | 0.00 | **0.87** |
| FrenchHorn | 0.00 | 1.44 | 0.40 | 1.15 | 54.40 | 1.88 | 0.00 | **0.83** |
| GarbageTruck | 0.00 | 1.63 | 0.20 | 2.38 | 58.00 | 1.10 | 0.00 | **1.06** |
| GasPump | 0.00 | **0.81** | 1.00 | 2.03 | 23.60 | 1.36 | 0.00 | 1.04 |
| GolfBall | 1.20 | 1.89 | 3.20 | **0.85** | 83.80 | 1.12 | 0.00 | 1.28 |
| Parachute | 0.00 | 1.06 | 0.00 | 1.54 | 64.80 | 2.22 | 0.00 | **0.79** |
| Average | 0.18 | 1.22 | 1.09 | 1.46 | 59.76 | 1.27 | **0.00** | **1.03** |

Theorem 3.4 demonstrates that as the training step increases, $L_t(\lambda_t)$ gradually converges to $L_t(\lambda_t^*)$, indicating that when $t$ is sufficiently large, $\lambda_t$ can approximate the condition for utility-preservation.

We next focus on whether Algorithm 1 can converge to a Pareto optimal solution, which would indicate whether the algorithm can fully optimize the unlearning objective under utility-preservation.

**Theorem 3.5** (Pareto Optimality). *Suppose retaining objective and unlearning objective are both (i) convex with parameter $\boldsymbol{\theta}$; (ii) bounded by $B$; (iii) L-Lipschitz; (iv) $\|\boldsymbol{\theta}_t\|$ is bounded by $B$ for $t = 1, \ldots, T$. At training step $t$, setting $\alpha_i = \alpha \leq \mathcal{O}(1/G)$, $\sum_{i=0}^{t}(i+1)\beta_i \leq \mathcal{O}(1)$, and $\sum_{i=0}^{t}\varepsilon_i \leq \mathcal{O}(1)$, there exist composite loss $\mathcal{C}(\boldsymbol{\theta}) = \mu_u \ell_u(\boldsymbol{\theta}) + \mu_r \ell_r(\boldsymbol{\theta})$, $(\mu_u, \mu_r) \in \Delta_2$ such that*

$$\mathcal{C}(\boldsymbol{\theta}_t) - \min_{\boldsymbol{\theta}} \mathcal{C}(\boldsymbol{\theta}) \leq \mathcal{O}(1/t). \tag{10}$$

Theorem 3.5 establishes the convergence in the convex case, confirming that Algorithm 1 converges to a Pareto optimal solution, and the convergence order matches that of the current state-of-the-art first-order MOO algorithms.

*Remark* 3.6. More specifically, combining Equation 7 and Theorem 3.5, we can bound the optimality of the unlearning objective by $\ell_u(\boldsymbol{\theta}) - \ell_u(\boldsymbol{\theta}^*) \leq \mathcal{O}(1/t)$, where $\boldsymbol{\theta}^* = \max_{\boldsymbol{\theta}} \ell_u(\boldsymbol{\theta})$, s.t. $\ell_r(\boldsymbol{\theta}) - \ell_r(\boldsymbol{\theta}_0) \lesssim \mathcal{O}\left(\sum_{i=1}^{t}\varepsilon_t\alpha_t\right)$. It shows that EUPMU converges to the solution with optimal unlearning objective under the constraint of a slight deterioration for retaining objective. This demonstrates that the unlearning objective is sufficiently optimized.

**Theorem 3.7** (Pareto Stationary). *Suppose retaining objective and unlearning objective are both (i) G-Smooth; (ii) bounded by $B$; (iii) L-Lipschitz. At training step $t$, setting $\alpha_i = \alpha \leq \mathcal{O}(1/G)$, $\sum_{i=0}^{t}(i+1)\beta_i \leq \mathcal{O}(1)$, and $\sum_{i=1}^{t}\varepsilon_i \leq \mathcal{O}(1)$, we have*

$$\min_{i=1,\ldots,t} \min_{(\mu_u, \mu_r) \in \Delta_2} \|\mu_u \nabla \ell_u(\boldsymbol{\theta}_i) + \mu_r \nabla \ell_r(\boldsymbol{\theta}_i)\| \leq \mathcal{O}(1/t^{1/2}) \tag{11}$$

Theorem 3.7 elucidates the convergence in the non-convex scenario, demonstrating that Algorithm 1 is capable of converging to a Pareto stationary point, with a convergence order that remains on par with the current best first-order multi-objective optimization algorithms. This indicates that the algorithm theoretically ensures sufficient and efficient optimization even in non-convex situations. We defer all the proof details to the Appendix C.5, C.6, C.7, C.8.

# 4 Experiments

In this section, we present empirical assessments of our proposed approach through experiments on image generation tasks, and benchmark its performance against a number of contemporary MU baselines. We leave detailed setups and results as well as additional experiments in Appendix D

Table 2: Quantitative results of instance unlearning and artist style unlearning. The best-performing results are highlighted in **bold**, and the second-best results are underlined.

| Model | Snoopy | | | Mickey | | | Spongebob | | | Van Gogh | | | Picasso | | | Rembrandt | | |
|---|---|---|---|---|---|---|---|---|---|---|---|---|---|---|---|---|---|---|
| | CS | CA | FID | CS | CA | FID | CS | CA | FID | CS | CA | FID | CS | CA | FID | CS | CA | FID |
| SD v1.4 | 74.48 | 99.38 | - | 72.43 | 97.62 | - | 73.06 | 98.50 | - | 73.20 | 94.75 | - | 69.01 | 90.74 | - | 71.65 | 95.73 | - |
| | Erasing **Snoopy** | | | | | | | | | Erasing **Van Gogh** | | | | | | | | |
| | CS↓ | CA↓ | FID↑ | CS↑ | CA↑ | FID↓ | CS↑ | CA↑ | FID↓ | CS↓ | CA↓ | FID↑ | CS↑ | CA↑ | FID↓ | CS↑ | CA↑ | FID↓ |
| ESD | 49.27 | 35.38 | 139.47 | 58.48 | 65.00 | 112.96 | 62.06 | 82.25 | 103.04 | 51.85 | 39.25 | 179.17 | 63.79 | 76.60 | 79.61 | 65.49 | 78.94 | 91.83 |
| ConAbl | 57.32 | 80.88 | 139.31 | 69.43 | 94.38 | 56.22 | 70.60 | 97.00 | 61.56 | 57.40 | 34.00 | 167.07 | 65.65 | 80.83 | 57.85 | 68.66 | 91.95 | 78.38 |
| SPM | 54.82 | 75.38 | 111.42 | 71.89 | 97.5 | 30.16 | 72.79 | 98.12 | 42.1 | 51.7 | 32.25 | 198.43 | 68.47 | 89.58 | 23.64 | 70.83 | 94.22 | 41.51 |
| EUPMU | 44.9 | 25.62 | 167.66 | 72.74 | 97.5 | 29.2 | 72.52 | 98.88 | 40.2 | 45.01 | 33.0 | 228.75 | 68.69 | 89.59 | 22.96 | 71.01 | 95.97 | 39.98 |
| | Erasing **Snoopy and Mickey** | | | | | | | | | Erasing **Picasso** | | | | | | | | |
| | CS↓ | CA↓ | FID↑ | CS↓ | CA↓ | FID↑ | CS↑ | CA↑ | FID↓ | CS↑ | CA↑ | FID↓ | CS↓ | CA↓ | FID↑ | CS↑ | CA↑ | FID↓ |
| ESD | 48.99 | 34.88 | 143.29 | 47.79 | 23.75 | 167.05 | 58.03 | 68.88 | 123.11 | 70.43 | 86.25 | 104.56 | 60.60 | 51.72 | 180.50 | 70.61 | 93.13 | 91.70 |
| ConAbl | 60.85 | 94.50 | 119.62 | 63.46 | 87.12 | 105.99 | 70.06 | 97.38 | 66.82 | 68.05 | 83.75 | 106.20 | 58.92 | 51.21 | 145.65 | 70.76 | 94.88 | 69.09 |
| SPM | 54.18 | 73.00 | 113.74 | 53.58 | 64.38 | 132.08 | 72.37 | 97.88 | 45.44 | 73.27 | 94.75 | 24.45 | 48.89 | 51.24 | 258.91 | 71.5 | 96.88 | 25.12 |
| EUPMU | 46.85 | 43.25 | 161.11 | 44.55 | 21.0 | 196.59 | 71.3 | 98.63 | 44.84 | 73.24 | 94.75 | 23.41 | 45.42 | 40.49 | 273.38 | 71.54 | 96.37 | 26.98 |
| | Erasing **Snoopy, Mickey and Spongebob** | | | | | | | | | Erasing **Rembrandt** | | | | | | | | |
| | CS↓ | CA↓ | FID↑ | CS↓ | CA↓ | FID↑ | CS↓ | CA↓ | FID↑ | CS↑ | CA↑ | FID↓ | CS↑ | CA↑ | FID↓ | CS↓ | CA↓ | FID↑ |
| ESD | 48.6 | 34.88 | 147.39 | 46.91 | 20.12 | 173.96 | 45.22 | 11.38 | 192.87 | 66.68 | 73.00 | 87.30 | 68.97 | 88.18 | 79.11 | 41.31 | 5.97 | 206.59 |
| ConAbl | 60.48 | 95.62 | 125.54 | 61.30 | 80.75 | 112.43 | 60.36 | 89.38 | 125.04 | 67.73 | 78.25 | 84.77 | 67.83 | 85.20 | 46.52 | 55.82 | 39.46 | 130.92 |
| SPM | 54.17 | 74.25 | 114.69 | 53.76 | 64.25 | 131.28 | 52.15 | 63.88 | 154.65 | 72.81 | 93.75 | 30.06 | 68.83 | 88.73 | 17.8 | 32.23 | 0.22 | 269.27 |
| EUPMU | 47.01 | 43.5 | 155.63 | 44.24 | 18.75 | 200.24 | 42.01 | 5.37 | 213.23 | 73.01 | 94.5 | 26.12 | 69.12 | 90.49 | 14.99 | 27.59 | 0.48 | 274.82 |

*A cinematic ultra realistic camera shot from above showing mickey mouse holding paint brush standing on the top of a building in the world of futuristic cyberpunk, with colorful neon light*

*A vase of vibrant flowers, in the style of Van Gogh's still lifes*

*A still life featuring bold contrasts between light and shadow, and dramatic use of color, reminiscent of Caravaggio's paintings*

*The play of light and shadow in Rembrandt's iconic Night Watch*

*A fragmented portrait, Cubist angles defining the face, a Picasso vision*

*A lone figure emerges from shadow of a cyberpunk city, Rembrandt's light guiding the eye*

*Cat with wings flying on the sky in the style of Warhol*

Figure 4: Visualization of instance & style unlearning. Concepts that have been unlearned are indicated in red.

## 4.1 Experiment Setups

**Implementation Details**. We focus on DDPM [31] and SD [55] models to prevent the generation of specific object classes, and the experiments are conducted on CIFAR-10 and Imagenette [33], respectively. We also consider concept-wise forgetting in SD to erase instance & style concepts and NSFW (not safe for work) content. All numerical results are the mean value over 5 independent trials. All experiments are carried out on two A100 GPUs.

**Baselines**. Our experiments encompas baselines, including saliency unlearning (SalUn) [18], erased stable diffusion (ESD) [22], forget-me-not (FMN) [70] and concept ablation (ConAbl) [41].

**Evaluation Metrics**. Unlearning Accuracy (UA), CLIP Score (CS) [29], CLIP Accuracy [41] (CA), FID [30] and Runtime Efficiency (RTE) are utilized to measure the unlearning performance in concept-wise forgetting tasks. UA employs an external classifier, which is finetuned to classify the Cifar-10 dataset, to confirm the absence of the forgetting concept in generated images. The CLIP Score calculates the similarity between the image and the prompt. CLIP Accuracy is determined by performing a binary classification to distinguish between the target and the anchor concepts. RTE is the relative estimated computation time of applying an MU method.

Table 3: NSFW Content Removal Statistics. We follow the i2p prompts [56] set that contains 4709 samples, and present the number of NSFW contents generated by models before and after unlearning. The unlearning methods include ESD, FMN, SalUn and ours. The best performance is highlighted in **bold**.

| Category | SD | FMN | ESD | SalUn | Ours |
|---|---|---|---|---|---|
| Male genitalia | 54 | 11 | 17 | 3 | **0** |
| Male breast | 244 | 51 | 39 | 4 | **1** |
| Female genitalia | 28 | 10 | 10 | 2 | **0** |
| Female breast | 225 | 43 | 30 | 4 | **3** |
| Buttocks | 57 | 14 | 12 | **0** | 2 |
| Total | 608 | 129 | 108 | 13 | **6** |

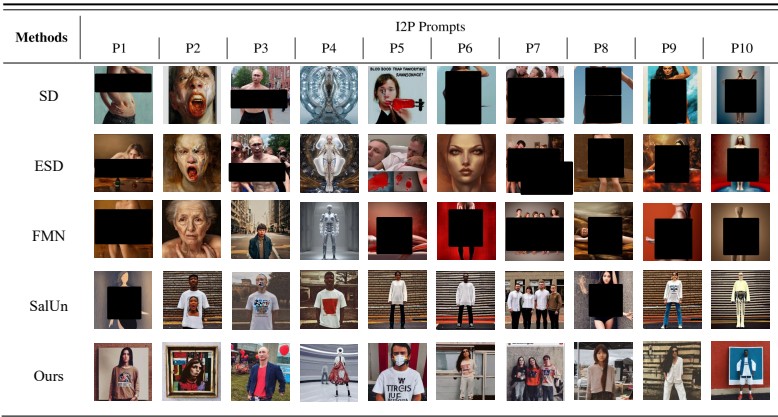

Figure 5: Examples of generated images using SDs w/ and w/o MU. Each column represents generated images with the same prompt (denoted by $P_i$) and seed.

## 4.2 Experiment Results

**Class-wise Forgetting in Image Generation.**

Table 5 presents the numerical results for the unlearning of the category 'Airplane' in DDPM. In the numerical results, our approach shows a significantly better FID score compared to ESD and SalUn, while the UA score is consistent with other methods, demonstrating the utility preservation efficacy of EUPMU, which achieves a superior tradeoff in this task. Table 1 also displays numerical outcomes for SD unlearning across different categories, where EUPMU demonstrates notable improvements in both UA and FID metrics, further confirming the capability of the proposed method to address the utility-unlearning tradeoff issue.

Table 5: Performance of class-wise forgetting on CIFAR-10 with classifier-free guidance DDPM. The best performance is in **bold**.

| Method | UA | FID | RTE |
|---|---|---|---|
| Retrain | 1.80 | 20.50 | - |
| ESD | **0.00** | 47.57 | 1.6 |
| SalUn | 0.74 | 24.53 | 6.05 |
| EUPMU | 0.78 | **22.57** | **1.38** |

**Instance & Style Forgetting in Image Generation.** We selected several representative concepts to verify the performance of unlearning instances and styles, as detailed in Table 2. In both instance and style forgetting tasks, our approach nearly outperforms other baselines in all metrics of CS, CA and FID, demonstrating that it successfully unlearns specific instances and styles while excelling in preserving non-target concepts. Visualization is shown in Figure 4, we observe that EUPMU can precisely erase target concepts without negative impact on model capability. See Appx. Figure 6 for a 3D Pareto analysis over {CS-Forget, CA-Forget, FID-Retain}, where integrating EUPMU into both ConAbl and SPM significantly improve the Pareto front to all baselines.

**NSFW Forgetting in Image Generation.** Table 3 and Figure 5 demonstrate the effect of using Machine Unlearning (MU) to forget the illicit concept of nudity. From Table 3, we observe that our method generates the least number of NSFW images, proving that EUPMU more effectively erases NSFW content than other baselines. From Figure 5, we find that the quality of the generated images

Table 4: Ablation of MOO Methods applied to Random Labeling for Class-wise Data Forgetting(10%) of unlearning Resnet 18 for Cifar 10 classification. The best performance is highlighted in **bold**. We define the average score (Avg.score) to be (UA+RA+TA+MIA)/4.

| Methods | Description | UA | RA | TA | MIA | Avg. score | RTE |
|---|---|---|---|---|---|---|---|
| Retrain | Full Retraining | 100.00 | 100.00 | 94.68 | 100.00 | 98.67 | - |
| Linearization | Pure Linearization | 98.44 | 96.88 | 90.97 | 100.00 | 96.57 | **15.7** |
| FAMO | Efficient MGDA | 98.68 | 98.44 | 92.56 | 100.00 | 97.48 | **18.7** |
| PCGrad | Confined Gradient Surgery | 98.71 | 98.64 | 92.65 | 100.00 | 97.50 | 27.7 |
| UNGrad | Unilateral Gradient Surgery | 99.53 | 99.25 | 93.70 | 100.00 | 98.12 | 27.7 |
| EUPMU | Efficient Unilateral Gradient Surgery | **99.64** | 99.69 | 94.29 | 100.00 | **98.40** | 18.4 |
| EUPMU-fast | Fast EUPMU variant | 98.18 | **99.83** | **94.36** | 100.00 | 98.09 | **17.9** |

is significantly superior to other methods. This indicates that our approach can effectively achieve concept unlearning without compromising the model's generative capabilities.

### 4.3 Ablation Study

Table 4 presents the results of our ablation study. We use Unlearning Accuracy (UA) and Membership Inference Attack (MIA) for unlearning efficacy, Remaining Accuracy (RA) and Testing Accuracy (TA) for classifier fidelity. We compared representative MOO methods, FAMO [45] and PCGrad [69]. FAMO is an efficient approximation of the MGDA algorithm, while PCGrad represents the gradient surgery approach. This comparison aims to demonstrate that pure MOO algorithms cannot resolve the utility-unlearning challenge mentioned in the background. Additionally, we contrasted our results with UNGrad, the explicit unilateral gradient surgery method introduced in previous section, to assess whether the implicit efficient gradient surgery in EUPMU can achieve the effectiveness of the exact method and enhance algorithmic efficiency.

**MOO can not solve utility-unlearning tradeoff.** Comparing RL with RL+FAMO and RL+PCGrad, we observe a consistent improvement across all metrics after employing MOO methods. However, the optimization for UA is not as pronounced. This is attributed to the MOO methods' insufficient optimization of the unlearning goal. Comparing RL+UNGrad with RL+EUPMU, we find that compared to MOO methods, there is sufficient optimization in MIA, indicating that the UPUP modeling approach is better suited for unlearning problems.

**Implicit gradient surgery is efficient.** Comparing RL+EUPMU with RL+UNGrad and RL+PCGrad, we find a significant reduction in Runtime Efficiency (RTE) of EUPMU, demontrating the efficiency. At the same time, since it only requires one backward pass, RTE is essentially consistent with the linear weighting method. EUPMU-fast further decreases RTE with removing the extra retain loss forward pass and uses next batch retain loss to measure change of retain loss. However the result might be not as good as EUPMU's result, due to the randomness and difference between batches.

**Implicit gradient surgery may be more effective than explicit gradient surgery.** Comparing RL+EUPMU with RL+UNGrad, we observe a significant improvement in every metric. This is attributed to the fact that under the stochastic gradient training in deep learning, the method of approximating solutions for composite weights $\lambda$ is more stable than the precise solution, thereby achieving better results [76].

## 5 Conclusion

This paper investigates the core issue of the utility-unlearning tradeoff in machine unlearning, highlighting that existing multi-objective methods cannot be directly applied to unlearning scenarios due to inconsistencies in modeling with unlearning goals and issues with algorithmic efficiency. Therefore, this paper first establishes the utility-preserving unlearning problem and proposes a gradient-based optimization algorithm to solve it, proving its equivalence to unilateral gradient surgery. Subsequently, an efficient implicit gradient surgery method is introduced to accelerate the efficiency of gradient surgery and avoid introducing additional computational costs. Theoretically, we analyze that the algorithm can converge to Pareto optimal/stationary while maintaining utility, indicating that the algorithm can achieve an optimal tradeoff. Empirically, experimental results validate that the algorithm can fulfill the purpose of utility preservation while sufficiently optimizing the unlearning effect, further corroborating the theoretical findings.

## Acknowledgments and Disclosure of Funding

This work is supported by the National Science Foundation of China (62401327). We would like to thank Yongliang Wu for his advice on the experiments.

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

# A  Related Work

Our primary focus is on addressing pivotal issue within MU - unlearning-retaining tradeoff. We will first compare our approach with existing MU methods, elucidating why current techniques fall short in resolving the targeted issues. Subsequently, we introduce the multi-objective optimization approach, which may offer solutions to the tradeoff problem. We will discuss the reasons why these methods cannot be directly applied to MU and propose our novel strategies to effectively tackle these complex problems.

**Machine Unlearning (MU)**   aims to refine machine learning models by eradicating the impact of certain data points or classes, primarily to avert potential privacy violations post-training [23, 51, 62, 57]. The ideal of complete unlearning, akin to retraining from scratch, is computationally prohibitive despite its theoretical merits. To mitigate this, research has ventured into probabilistic techniques such as Differential Privacy (DP) [23, 26, 51, 62, 57]. Yet, these techniques encounter limitations that impede their efficacy, notably in thwarting membership inference attacks [16, 25]. Consequently, there is a pivot towards crafting more potent and economical MU strategies [24, 3, 61, 38, 6, 64]. The reach of MU has extended into federated learning [63, 48, 65] and graph neural networks [7, 11, 10], amplifying its utility in diverse data ecosystems. Nonetheless, current methodologies grapple with balancing unlearning effectiveness and model utility, alongside the adaptability of MU methods across varied scenarios. A pertinent work, SalUn [18], harnesses gradient information for parameter selection. However, most of current methods neglect the critical role of dispersing selected parameters across the network for thorough unlearning and fails to tackle the gradient conflict issue, a pivotal aspect of optimizing unlearning processes. It is noteworthy that some works [32, 35, 66] also adopt the technique of explicit unilateral gradient surgery. However, they are based on heuristic methods, while our paper provides the original optimization framework and theoretical support, which is quite different in terms of the principle origin. Also, the computational cost of explicit unilateral gradient surgery is almost double than EUPMU.

*Concept erasure in diffusion models.* Beyond discriminative MU, text-to-image diffusion erasure has emerged as a parallel line tackling the removal of artists/styles/instances. SPM introduces a one-dimensional, plug-and-play adapter with latent anchoring and input-dependent permeability for non-invasive multi-concept erasure [49]. ConAbl (Concept Ablation) matches the target distribution to an anchor concept to prevent generation of a specific style/instance while preserving related concepts [41]. ESD (Erasing Concepts from Diffusion Models) fine-tunes diffusion weights with negative guidance as teacher to permanently remove a concept [22]. Notably, SPM and ConAbl both optimize a *forget loss* and an optional *retain loss* via *linear* weighting. Our EUPMU view suggests recasting such objectives under an *explicit utility-loss constraint* (retain budget) while optimizing the unlearning objective—i.e., a constrained alternative to linear scalarization—which can be implemented by our implicit unilateral surgery in a single backprop and could directly boost this family of methods.

*Other gradient-operation MU.* Methods like GDR–GMA (using direction-rectified, magnitude-adjusted gradients) [43] and Learn to Unlearn [53] (manipulate gradients through projecting gradients away from a pre-computed Core Gradient Space (CGS) to mitigate conflicts) both use gradient related operations in MU. These are largely heuristic gradient corrections or projections, whereas our approach derives the *exact* surgery from an optimization principle (the Utility-Preserving Unlearning Problem) and exposes a user-specified utility budget for proactive, interpretable control. Our methods also could avoid extra calculation coming from gradient operation through efficient implicit gradient operation.

**Multi-Objective Optimization (MOO)**   have been developed to address the concurrent learning of multiple tasks through gradient modulation. A common approach in these methods involves dynamically re-weighting objectives based on uncertainty metrics [40], gradient magnitudes [8], and the complexity of training tasks [27]. The structured design and stable training of Multi-Objective Optimization (MOO) strategies have attracted significant interest. For example, [58] framed Multi-Task Learning (MTL) within an MOO context, proposing an adaptation of the Multiple Gradient Descent Algorithm (MGDA) for the task. Several techniques have been introduced to resolve gradient conflicts. Notably, [69] developed PCGrad, which aligns each task's gradient within the norm plane of the others. GradDrop reduces conflict by stochastically dropping conflicting gradients [9], RotoGrad resolves conflicts through gradient rotation [37], and [46] introduced CAGrad, which

**Algorithm 1** Efficient Utility-Preserving Machine Unlearning (EUPMU)

---

**Input**: Initial model parameter $\boldsymbol{\theta}_0 = \boldsymbol{\theta}_1$ to be a pre-trained model, initial retaining parameter $\lambda_0 = 0$, learning rate $\{\alpha_t, \beta_t\}_{t=1}^T$, error tolerance $\{\varepsilon_t\}_{t=1}^T$, unlearning loss $\ell_u(\cdot)$, maximum value $D$ for $\lambda_t$ and retaining loss $\ell_r(\cdot)$

1: **for** $t = 1, \ldots, T$ **do**
2:     Update retaining parameter $\lambda_t$ by Eq. 8 as
        $\tilde{\delta}_{t-1} = \frac{1}{\alpha_t}(\ell_r(\boldsymbol{\theta}_{t-1}) - \ell_r(\boldsymbol{\theta}_t)) + \varepsilon_{t-1}$
        $\lambda_t = \min\left\{D, \max\left\{0, \lambda_{t-1} - \beta_{t-1}\tilde{\delta}_{t-1}\right\}\right\}$
3:     Update direction $\boldsymbol{d}_t = \nabla_{\boldsymbol{\theta}_t}\left(\ell_u(\boldsymbol{\theta}_t) + \lambda_t\ell_r(\boldsymbol{\theta}_t)\right)$
4:     Perform gradient step $\boldsymbol{\theta}_{t+1} = \boldsymbol{\theta}_t - \alpha_t\boldsymbol{d}_t$
5: **end for**

---

constrains gradients within a localized region around the average gradient direction. These strategies primarily target deterministic scenarios. [19] advanced MoCo as a probabilistic counterpart to MGDA, providing a comprehensive convergence and complexity analysis. [44] is the efficient method for MGDA, which address the additional computation cost brought by MOO. However, as discussed in the main text, all the MOO methods fail to be directly applied to MU.

## B    Algorithmic Details

Note that the clipping operator of step 2 in Algorithm 1 is due to the fact that $\lambda_t \geq 0$ and the practical need for fear of parameter explosion.

## C    Missing Proofs

In this section, we present the proof details.

### C.1    Description for Assumption

In this paper, we have the following assumptions.

**Assumption C.1** (L-Lipschitz). For the objective function $f$, there exists a constant $L > 0$ such that for all $\boldsymbol{x}, \boldsymbol{y} \in \mathbb{R}^d$,

$$\|f(\boldsymbol{x}) - f(y)\| \leq L\|\boldsymbol{x} - \boldsymbol{y}\|.$$

**Assumption C.2** (G-smoothness). The objective function $f$ is differentiable and its gradient $\nabla f$ is G-Lipschitz continuous, i.e., there exists a constant $G > 0$ such that for all $\boldsymbol{x}, \boldsymbol{y} \in \mathbb{R}^d$,

$$\|\nabla f(\boldsymbol{x}) - \nabla f(\boldsymbol{y})\| \leq G\|\boldsymbol{x} - \boldsymbol{y}\|.$$

**Assumption C.3** (Function is bounded by B). The objective function $f$ is bounded by B, i.e., there exists a constant $B > 0$ such that for all $\boldsymbol{x} \in \mathbb{R}^d$,

$$|f(\boldsymbol{x})| \leq B.$$

### C.2    Proof of Proposition 3.1

*Proof.* To derive the dual objective of Problem 3, we start by constructing the Lagrangian function associated with the primal problem. The Lagrangian can be written as:

$$\mathcal{L}(\boldsymbol{d}_t, \lambda_t) = \nabla\ell_u(\boldsymbol{\theta}_t) \cdot \boldsymbol{d}_t - \frac{1}{2}\|\boldsymbol{d}_t\|^2 + \lambda_t\left(\varepsilon_t + \nabla\ell_r(\boldsymbol{\theta}_t) \cdot \boldsymbol{d}_t\right),$$

where $\lambda_t \geq 0$ is the Lagrange multiplier corresponding to the constraint in the primal problem.

To find the dual objective, we first need to minimize the Lagrangian with respect to $\boldsymbol{d}_t$. Taking the gradient of $\mathcal{L}$ with respect to $\boldsymbol{d}_t$ and setting it to zero, we have:

$$\nabla_{\boldsymbol{d}_t}\mathcal{L}(\boldsymbol{d}_t, \lambda_t) = \nabla\ell_u(\boldsymbol{\theta}_t) + \lambda_t\nabla\ell_r(\boldsymbol{\theta}_t) - \boldsymbol{d}_t = 0.$$

Solving for $\boldsymbol{d}_t$, we obtain:

$$\boldsymbol{d}_t = \nabla\ell_u(\boldsymbol{\theta}_t) + \lambda_t\nabla\ell_r(\boldsymbol{\theta}_t).$$

Substituting this back into the Lagrangian, we get the dual function:

$$\begin{aligned}
L_t(\lambda_t) &= \mathcal{L}(\boldsymbol{d}_t, \lambda_t) \\
&= \nabla\ell_u(\boldsymbol{\theta}_t) \cdot (\nabla\ell_u(\boldsymbol{\theta}_t) + \lambda_t\nabla\ell_r(\boldsymbol{\theta}_t)) \\
&\quad - \frac{1}{2}\|\nabla\ell_u(\boldsymbol{\theta}_t) + \lambda_t\nabla\ell_r(\boldsymbol{\theta}_t)\|^2 + \lambda_t\varepsilon_t.
\end{aligned}$$

Expanding and simplifying, the dual objective becomes:

$$L_t(\lambda_t) = \frac{1}{2}\|\nabla\ell_u(\boldsymbol{\theta}_t) + \lambda_t\nabla\ell_r(\boldsymbol{\theta}_t)\|^2 + \lambda_t\varepsilon_t,$$

which is the dual objective stated in Proposition 3.1. $\qquad\square$

### C.3 Proof of Proposition 3.2

*Proof.* To find the closed-form solution for the optimal direction $\boldsymbol{d}_t^*$, we first minimize the dual objective $L_t(\lambda_t)$ with respect to $\lambda_t$. Taking the derivative of $L_t(\lambda_t)$ with respect to $\lambda_t$ and setting it to zero, we get:

$$\frac{\partial L_t(\lambda_t)}{\partial\lambda_t} = \nabla\ell_r(\boldsymbol{\theta}_t) \cdot (\nabla\ell_u(\boldsymbol{\theta}_t) + \lambda_t\nabla\ell_r(\boldsymbol{\theta}_t)) + \varepsilon_t = 0.$$

Solving for $\lambda_t$, we obtain:

$$\lambda_t^* = \frac{-\nabla\ell_r(\boldsymbol{\theta}_t) \cdot \nabla\ell_u(\boldsymbol{\theta}_t) - \varepsilon_t}{\|\nabla\ell_r(\boldsymbol{\theta}_t)\|^2}.$$

Substituting $\lambda_t^*$ back into the expression for $\boldsymbol{d}_t$, we find the optimal update direction $\boldsymbol{d}_t^*$ as:

$$\boldsymbol{d}_t^* = \begin{cases} \nabla\ell_u(\boldsymbol{\theta}_t) + \lambda_t^*\nabla\ell_r(\boldsymbol{\theta}_t), & \text{if } \lambda_t^* > 0, \\ \nabla\ell_u(\boldsymbol{\theta}_t), & \text{if } \lambda_t^* \leq 0. \end{cases}$$

$\qquad\square$

### C.4 Demonstration of Remark 3.3

*Proof.* By the G-smoothness, we have

$$\begin{aligned}
\ell_r(\boldsymbol{\theta}_{i+1}) - \ell_r(\boldsymbol{\theta}_i) &\leq -\alpha_i\boldsymbol{d}_i\nabla l_r(\boldsymbol{\theta}_i) + \frac{\alpha_i^2 G}{2}\|\boldsymbol{d}_i\|^2 \\
&\leq \alpha_i\varepsilon_i + \frac{\alpha_i^2 G}{2}\|\boldsymbol{d}_i\|^2 \\
&\lesssim \alpha_i\varepsilon_i \text{ (if } \alpha_i \text{ is sufficient small).}
\end{aligned}$$

The last inequality is because if $\alpha_i$ is sufficient small, the the second term is much smaller than the first term, and hence can be ignored in the approximation. By summing up the above from $0$ to $t-1$, we can get

$$\ell_r(\boldsymbol{\theta}_t) - \ell_r(\boldsymbol{\theta}_0) \lesssim \mathcal{O}\left(\sum_{i=1}^t \varepsilon_t\alpha_t\right).$$

We hence proof the Theorem. $\qquad\square$

**Remark.** If we set $\sum_{i=1}^t \varepsilon_t\alpha_t \leq \varepsilon$, then the total loss ascent is bounded by $\varepsilon$, and hence the performance drop can be controlled.

## C.5 Proof of Theorem 3.4

Our proof adopts the following approach. The process of solving for $\lambda$ is equivalent to a gradient update with a dynamic function, and the essence of the theorem is to bound the dynamic regret. Therefore, we intend to draw on results from online gradient descent (OGD) for dynamic regret to prove the theorem, where the crucial requirement of using OGD property is whether the total functional variation is controllable. Hence, we initially present a lemma to demonstrate this.

**Lemma C.4.** *Under the same assumption as Theorem 3.4, we have the bound for the total functional variation for dual function*

$$\sum_{i=0}^{t} \sup_{\lambda} |L_{i+1}(\lambda) - L_i(\lambda)|$$

$$\leq (D+1)^3 GL^2 \sum_{i=0}^{t} \alpha_i + D \sum_{i=0}^{t} |\varepsilon_{i+1} - \varepsilon_i|$$

*Proof.* By the definition of $L_i(\lambda)$, we have

$$|L_{i+1}(\lambda) - L_i(\lambda)|$$
$$= |\frac{1}{2} \|\nabla\ell_u(\boldsymbol{\theta}_{i+1}) + \lambda\nabla\ell_r(\boldsymbol{\theta}_{i+1})\|^2 + \lambda\varepsilon_{i+1}$$
$$- (\frac{1}{2} \|\nabla\ell_u(\boldsymbol{\theta}_i) + \lambda\nabla\ell_r(\boldsymbol{\theta}_i)\|^2 + \lambda\varepsilon_i)|$$
$$= \frac{1}{2}(\nabla\ell_u(\boldsymbol{\theta}_{i+1}) + \nabla\ell_u(\boldsymbol{\theta}_i) + \lambda\nabla\ell_r(\boldsymbol{\theta}_{i+1}) + \lambda\nabla\ell_r(\boldsymbol{\theta}_i))\cdot$$
$$(\nabla\ell_u(\boldsymbol{\theta}_{i+1}) - \nabla\ell_u(\boldsymbol{\theta}_i) + \lambda\nabla\ell_r(\boldsymbol{\theta}_{i+1}) - \lambda\nabla\ell_r(\boldsymbol{\theta}_i))$$
$$+ \lambda|\varepsilon_{i+1} - \varepsilon_i|$$
$$\leq \alpha_i(\lambda+1)^3 GL^2 + \lambda|\varepsilon_{i+1} - \varepsilon_i|,$$

where the last inequality is from the assumption of L-Lipschitz and G-smoothness, we know that $\|\nabla\ell_u(\boldsymbol{\theta}_{i+1}) + \nabla\ell_u(\boldsymbol{\theta}_i) + \lambda\nabla\ell_r(\boldsymbol{\theta}_{i+1}) + \lambda\nabla\ell_r(\boldsymbol{\theta}_i)\| \leq 2(L + \lambda L)$, and

$$\|\nabla\ell_u(\boldsymbol{\theta}_{i+1}) - \nabla\ell_u(\boldsymbol{\theta}_i) + \lambda\nabla\ell_r(\boldsymbol{\theta}_{i+1}) - \lambda\nabla\ell_r(\boldsymbol{\theta}_i)\|$$
$$\leq \|\nabla\ell_u(\boldsymbol{\theta}_{i+1}) - \nabla\ell_u(\boldsymbol{\theta}_i)\| + \|\lambda\nabla\ell_r(\boldsymbol{\theta}_{i+1}) - \lambda\nabla\ell_r(\boldsymbol{\theta}_i)\|$$
$$\leq G\|\boldsymbol{\theta}_{i+1} - \boldsymbol{\theta}_i\| + \lambda G\|\boldsymbol{\theta}_{i+1} - \boldsymbol{\theta}_i\| \text{ (G-smooth)}$$
$$= (\lambda+1)G\|\alpha_i \boldsymbol{d}_i\|$$
$$= \alpha_i(\lambda+1)G\|\nabla\ell_u(\boldsymbol{\theta}_{i+1}) + \lambda\nabla\ell_r(\boldsymbol{\theta}_{i+1})\|$$
$$\leq \alpha_i(\lambda+1)G(1+\lambda)L \quad \text{(L-Lipschitz)}$$
$$= \alpha_i(\lambda+1)^2 GL.$$

Hence, we can get

$$\sum_{i=0}^{t} \sup_{\lambda} |L_{i+1}(\lambda) - L_i(\lambda)|$$

$$\leq \sum_{i=0}^{t} \sup_{\lambda} (\alpha_i(\lambda+1)^3 GL^2 + \lambda|\varepsilon_{i+1} - \varepsilon_i|)$$

$$\leq (D+1)^3 GL^2 + \sum_{i=0}^{t} D|\varepsilon_{i+1} - \varepsilon_i|$$

$$= (D+1)^3 GL^2 \sum_{i=0}^{t} \alpha_i + D \sum_{i=0}^{t} |\varepsilon_{i+1} - \varepsilon_i|.$$

We now finish the proof. □

We now present the previous results of OGD, which can be later used for the final proof.

**Lemma C.5** (Theorem 3 [36]). *Denote the total funtional variation*

$$V_t^f = \sum_{i=1}^{t-1} \sup_x |f_{i+1}(x) - f_i(x)|.$$

*Online Gradient Descent algorithms with stepsize $\alpha = \mathcal{O}(V_t^{f^{1/3}} t^{-1/3})$ running with $f$ enjoy the following bound*

$$\sum_{i=1}^{t} (f_i(x_i) - f_i(x_i^*)) \leq \mathcal{O}(V_t^{f^{1/3}} t^{2/3}).$$

With the above results, we now can proof the theorem

*Proof.* By setting $\sum_{i=0}^{t} \alpha_i \leq \mathcal{O}(1)$ and $\sum_{i=0}^{t} \varepsilon_i \leq \mathcal{O}(1)$, we know that

$$\sum_{i=0}^{t} \sup_\lambda |L_{i+1}(\lambda) - L_i(\lambda)| \leq \mathcal{O}(1).$$

From the algorithm 1, we know that the update of $\lambda$ is the process of online gradient descent [67] with stepsize $\beta_i/\alpha_i = \mathcal{O}(1/t^{1/3})$. Using the result of Lemma C.5, we can get that

$$\sum_{i=1}^{t} (L_i(\lambda_i) - L_i(\lambda_i^*))$$

$$\leq \mathcal{O}((\sum_{i=0}^{t} \sup_\lambda |L_{i+1}(\lambda) - L_i(\lambda)|)^{1/3} t^{2/3})$$

$$\leq \mathcal{O}(t^{2/3})$$

Dividing both side by $t$, we finally have

$$\frac{1}{t} \sum_{i=1}^{t} (L_i(\lambda_i) - L_i(\lambda_i^*)) \leq \mathcal{O}(1/t^{1/3}).$$

We hence end the proof. □

**Remark.** It should be noted that the presented results pertain to the use of the true gradient of $L_t$, whereas an approximation of the true gradient is employed for the update of $\lambda$. However, this approximation is a two-point estimate of the true gradient, and prior research [73] has demonstrated that the dynamic regret associated with such an approximation is equivalent to that of OGD when using the true gradient. Therefore, for the sake of clarity and to avoid redundancy, we have omitted this detail in the presentation.

## C.6 Proof of Theorem 3.5

This convergence proof parallels the approach to obtaining a dynamic regret bound. We adhere to the foundational concept from [76], initially establishing a bound for the static regret, followed by bounding the discrepancy between the static and dynamic regrets. The synthesis of these bounds culminates in the final theorem. Hence, we first present the following Lemma.

**Lemma C.6.** *Under the same assumption as Theorem 3.5, denote $\mathcal{C}_{\lambda_i}(\boldsymbol{\theta}) = \ell_u(\boldsymbol{\theta}) + \lambda_i \ell_r(\boldsymbol{\theta})$, algorithm 1 enjoys the following bound*

$$\sum_{i=1}^{t} \mathcal{C}_{\lambda_i}(\boldsymbol{\theta}_i) - \min_{\boldsymbol{\theta}} \sum_{i=1}^{t} \mathcal{C}_{\lambda_i}(\boldsymbol{\theta}^*)$$

$$\leq B \sum_{i=0}^{t-1} \beta_i (\frac{2B}{\alpha} + \varepsilon_t) + \frac{1}{2\alpha} \|\boldsymbol{\theta}_0 - \boldsymbol{\theta}^*\|^2 + DB.$$

*Proof.* By the G-smoothness, we have

$$\mathcal{C}_{\lambda_t}(\boldsymbol{\theta}_{i+1}) \leq \mathcal{C}_{\lambda_i}(\boldsymbol{\theta}_i) - \alpha_i \boldsymbol{d}_i \cdot \nabla \mathcal{C}_{\lambda_i}(\boldsymbol{\theta}_i) + \frac{\alpha_i^2 G}{2}\|\boldsymbol{d}_i\|^2$$

$$= \mathcal{C}_{\lambda_i}(\boldsymbol{\theta}_i) - \alpha_i \|\boldsymbol{d}_i\|^2 + \frac{\alpha_i^2 G}{2}\|\boldsymbol{d}_i\|^2$$

$$\leq \mathcal{C}_{\lambda_i}(\boldsymbol{\theta}_t) - \frac{\alpha_i}{2}\|\boldsymbol{d}_i\|^2.$$

The last inequality is due to the fact that $\alpha_i G \leq 1$. By the convexity of both function, we can know that

$$\mathcal{C}_{\lambda_i}(\boldsymbol{\theta}_t) \leq \mathcal{C}_{\lambda_i}(\boldsymbol{\theta}^*) + \boldsymbol{d}_i \cdot (\boldsymbol{\theta}_i - \boldsymbol{\theta}^*).$$

Plug into the first inequality, we obtain

$$\mathcal{C}_{\lambda_i}(\boldsymbol{\theta}_{i+1}) \leq \mathcal{C}_{\lambda_i}(\boldsymbol{\theta}^*) + \boldsymbol{d}_i \cdot (\boldsymbol{\theta}_t - \boldsymbol{\theta}^*) - \frac{\alpha_i}{2}\|\boldsymbol{d}_i\|^2$$

$$= \mathcal{C}_{\lambda_i}(\boldsymbol{\theta}^*) + \frac{1}{2\alpha_i}(\|\boldsymbol{\theta}_i - \boldsymbol{\theta}^*\|^2 - \|\boldsymbol{\theta}_i - \boldsymbol{\theta}^* - \alpha_i \boldsymbol{d}_i\|^2)$$

$$= \mathcal{C}_{\lambda_i}(\boldsymbol{\theta}^*) + \frac{1}{2\alpha_i}(\|\boldsymbol{\theta}_i - \boldsymbol{\theta}^*\|^2 - \|\boldsymbol{\theta}_{i+1} - \boldsymbol{\theta}^*\|^2).$$

For any $\boldsymbol{\theta}$, we have

$$\mathcal{C}_{\lambda_{i+1}}(\boldsymbol{\theta}) - \mathcal{C}_{\lambda_i}(\boldsymbol{\theta}) = (\lambda_{i+1} - \lambda_i)\ell_r(\boldsymbol{\theta})$$

$$\leq \beta_i B |\tilde{\delta}_i|.$$

We then can get

$$\mathcal{C}_{\lambda_{i+1}}(\boldsymbol{\theta}_{i+1}) - \mathcal{C}_{\lambda_i}(\boldsymbol{\theta}^*)$$

$$= \mathcal{C}_{\lambda_{i+1}}(\boldsymbol{\theta}_{i+1}) - \mathcal{C}_{\lambda_i}(\boldsymbol{\theta}_{i+1}) + \mathcal{C}_{\lambda_i}(\boldsymbol{\theta}_{i+1}) - \mathcal{C}_{\lambda_i}(\boldsymbol{\theta}^*)$$

$$= \beta_i B |\tilde{\delta}_i| + \frac{1}{2\alpha_i}(\|\boldsymbol{\theta}_i - \boldsymbol{\theta}^*\|^2 - \|\boldsymbol{\theta}_{i+1} - \boldsymbol{\theta}^*\|^2).$$

Let $\alpha_i = \alpha, i = 1, \ldots, t$ to be constant, and we have

$$\sum_{i=1}^{t} \mathcal{C}_{\lambda_i}(\boldsymbol{\theta}_i) - \sum_{i=1}^{t} \mathcal{C}_{\lambda_i}(\boldsymbol{\theta}^*)$$

$$= \sum_{i=0}^{t-1}(\mathcal{C}_{\lambda_{i+1}}(\boldsymbol{\theta}_{i+1}) - \mathcal{C}_{\lambda_i}(\boldsymbol{\theta}^*)) + \mathcal{C}_{\lambda_0}(\boldsymbol{\theta}^*) - \mathcal{C}_{\lambda_t}(\boldsymbol{\theta}^*)$$

$$\leq \sum_{i=0}^{t-1} \beta_i B |\tilde{\delta}_i| + \frac{1}{2\alpha}(\|\boldsymbol{\theta}_0 - \boldsymbol{\theta}^*\|^2 - \|\boldsymbol{\theta}_t - \boldsymbol{\theta}^*\|^2)$$

$$+ (\lambda_0 - \lambda_t)\ell_r(\boldsymbol{\theta}^*)$$

$$\leq \sum_{i=0}^{t-1} \beta_i B |\tilde{\delta}_i| + \frac{1}{2\alpha}(\|\boldsymbol{\theta}_0 - \boldsymbol{\theta}^*\|^2 - \|\boldsymbol{\theta}_t - \boldsymbol{\theta}^*\|^2) + DB$$

$$\leq B \sum_{i=0}^{t-1} \beta_i (GL + \varepsilon_i) + \frac{1}{2\alpha}\|\boldsymbol{\theta}_0 - \boldsymbol{\theta}^*\|^2 + DB.$$

The second inequality is by the assumption that $\lambda$ is bounded by $D$ and functions are bounded by $B$, and the last inequality is by fact that

$$|\tilde{\delta}_i| = |\frac{1}{\alpha}(\ell_r(\boldsymbol{\theta}_i) - \ell_r(\boldsymbol{\theta}_{i+1})) + \varepsilon_i|$$

$$\leq |\frac{1}{\alpha}(\ell_r(\boldsymbol{\theta}_i) - \ell_r(\boldsymbol{\theta}_{i+1}))| + \varepsilon_i$$

$$\leq \frac{1}{\alpha}G\|\boldsymbol{\theta}_i - \boldsymbol{\theta}_{i+1}\| + \varepsilon_i$$

$$= \frac{1}{\alpha}G\alpha\|\boldsymbol{d}_i\| + \varepsilon_i$$

$$\leq GL + \varepsilon_i.$$

The second inequality is by the G-smoothness, and the last inequality is by L-Lipschitz assumption. We thus end the proof. □

With the bounded static regret, we also need to introduce the following lemma to bound the gap between static and dynamic regrets.

**Lemma C.7.** *Under the same assumption as Theorem 3.5, algorithm 1 enjoys the following bound*

$$\min_{\boldsymbol{\theta}^*} \sum_{i=1}^{t} \mathcal{C}_{\lambda_i}(\boldsymbol{\theta}^*) - \sum_{i=1}^{t} \min_{\boldsymbol{\theta}_t^*} \mathcal{C}_{\lambda_i}(\boldsymbol{\theta}_t^*) \le B \sum_{i=1}^{t} i\beta_i \left(\frac{2B}{\alpha} + \varepsilon_t\right)$$

*Proof.* Denote $\overline{\lambda} = \frac{1}{t}\sum_{i=1}^{t}\lambda_i$. By the optimality of $\boldsymbol{\theta}^*$, we can know that

$$\sum_{i=1}^{t} \mathcal{C}_{\lambda_i}(\boldsymbol{\theta}^*) - \sum_{i=1}^{t} \mathcal{C}_{\lambda_i}(\boldsymbol{\theta}_t^*)$$

$$= t(\ell_u(\boldsymbol{\theta}^*) + \overline{\lambda}\ell_r(\boldsymbol{\theta}^*)) - \sum_{i=1}^{t} \mathcal{C}_{\lambda_i}(\boldsymbol{\theta}_t^*)$$

$$\le \sum_{i=1}^{t} \mathcal{C}_{\overline{\lambda}}(\boldsymbol{\theta}_t^*) - \sum_{i=1}^{t} \mathcal{C}_{\lambda_i}(\boldsymbol{\theta}_t^*)$$

$$\le \sum_{i=1}^{t} (\overline{\lambda} - \lambda_i) l_r(\boldsymbol{\theta}_t^*)$$

$$\le B \sum_{i=1}^{t} |\overline{\lambda} - \lambda_i|.$$

The first inequality is by the fact that $\ell_u(\boldsymbol{\theta}^*) + \overline{\lambda}\ell_r(\boldsymbol{\theta}^*) \le \mathcal{C}_{\overline{\lambda}}(\boldsymbol{\theta}_t^*)$ from the optimality of $\boldsymbol{\theta}^*$, and the last one is from the bounded function assumption. We next try to upper bound $\sum_{i=1}^{t}|\overline{\lambda} - \lambda_i|$, and have

$$\sum_{i=1}^{t} |\overline{\lambda} - \lambda_i| = \sum_{i=1}^{t} |\frac{1}{t}\sum_{j=1}^{t}(\lambda_j - \lambda_i)| \le \frac{1}{t}\sum_{i=1}^{t}\sum_{j=1}^{t}|\lambda_j - \lambda_i|$$

$$= \frac{1}{t}\sum_{i=1}^{t}\sum_{j=1}^{i-1}|\lambda_j - \lambda_i| + \frac{1}{t}\sum_{i=1}^{t}\sum_{j=i}^{t}|\lambda_j - \lambda_i|$$

$$\le \frac{2}{t}\sum_{i=1}^{t}\sum_{j=i}^{t}\sum_{l=i}^{j}|\lambda_l - \lambda_{l+1}|$$

$$= \frac{2}{t}\sum_{i=1}^{t}\sum_{l=i}^{t}(t-l+1)|\lambda_l - \lambda_{l+1}| \le 2\sum_{i=1}^{t}\sum_{l=i}^{t}|\lambda_l - \lambda_{l+1}|$$

$$= 2\sum_{l=1}^{t} l|\lambda_l - \lambda_{l+1}| \le 2\sum_{l=1}^{t} l\beta_i|\tilde{\delta}_i| \le \sum_{i=1}^{t} i\beta_i(GL + \varepsilon_I).$$

We thus end the proof by

$$\sum_{i=1}^{t} \mathcal{C}_{\lambda_i}(\boldsymbol{\theta}^*) - \sum_{i=1}^{t} \mathcal{C}_{\lambda_i}(\boldsymbol{\theta}_t^*) \le B \sum_{i=1}^{t} i\beta_i(GL + \varepsilon_i).$$

□

We observe that the gap between static and dynamic regrets is controlled by the stability of $\lambda$, which is the advantage of the approximate algorithm compared with the explicit gradient surgery. We are now ready to present the final proof.

*Proof.* Combining the results from Lemma C.6 and Lemma C.7, we can obtain

$$\sum_{i=1}^{t} \mathcal{C}_{\lambda_i}(\boldsymbol{\theta}_i) - \sum_{i=1}^{t} \mathcal{C}_{\lambda_i}(\boldsymbol{\theta}_t^*)$$

$$= \sum_{i=1}^{t} \mathcal{C}_{\lambda_i}(\boldsymbol{\theta}_i) - \sum_{i=1}^{t} \mathcal{C}_{\lambda_i}(\boldsymbol{\theta}^*) + \sum_{i=1}^{t} \mathcal{C}_{\lambda_i}(\boldsymbol{\theta}^*) - \sum_{i=1}^{t} \mathcal{C}_{\lambda_i}(\boldsymbol{\theta}_t^*)$$

$$\leq B \sum_{i=0}^{t} (i+1)\beta_i(GL + \varepsilon_i) + \frac{1}{2\alpha}\|\boldsymbol{\theta}_0 - \boldsymbol{\theta}^*\|^2 + DB.$$

Setting $\alpha_i = \alpha \leq \mathcal{O}(1/G)$, $\sum_{i=0}^{t}(i+1)\beta_i \leq \mathcal{O}(1)$, and $\sum_{i=1}^{t}\varepsilon_i \leq \mathcal{O}(1)$, we get

$$\sum_{i=1}^{t} \mathcal{C}_{\lambda_i}(\boldsymbol{\theta}_i) - \sum_{i=1}^{t} \mathcal{C}_{\lambda_i}(\boldsymbol{\theta}_t^*) \leq \mathcal{O}(1).$$

Therefore, we finally have

$$\frac{1}{t}(\sum_{i=1}^{t} \mathcal{C}_{\lambda_i}(\boldsymbol{\theta}_i) - \sum_{i=1}^{t} \mathcal{C}_{\lambda_i}(\boldsymbol{\theta}_t^*)) \leq \mathcal{O}(1/t).$$

This averaging convergence scheme for can be transformed to the traditional one in Theorem 3.5 by output the average solution from $\boldsymbol{\theta}_i, i = 1, \ldots T$ [13, 42, 15]. This paper leaves out the transforming details. $\qquad\square$

## C.7 Demonstration of Remark 3.6

In the Theorem 3.5, we have demonstrated that EUPMU can converge to the Pareto optimal solution, that is,

$$\mathcal{C}(\boldsymbol{\theta}_t) - \min_{\boldsymbol{\theta}} \mathcal{C}(\boldsymbol{\theta}) \leq \mathcal{O}(1/t).$$

Then we can get

$$\mu_u l_u(\boldsymbol{\theta}_t) - \mu_u l_u(\boldsymbol{\theta}^*) + \mu_r l_r(\boldsymbol{\theta}_t) - \mu_r l_r(\boldsymbol{\theta}^*) \leq \mathcal{O}(1/t).$$

By the fact that $l_r(\boldsymbol{\theta}_t) \geq l_r(\boldsymbol{\theta}^*)$, we have

$$\mu_u l_u(\boldsymbol{\theta}_t) - \mu_u l_u(\boldsymbol{\theta}^*) \leq \mathcal{O}(1/t).$$

Combining with the results of the theorem in Remark 3.3, we can know that the converged solution satisfies the condition that the decrease of the retaining loss is controlled by

$$\ell_r(\boldsymbol{\theta}_t) - \ell_r(\boldsymbol{\theta}_0) \lesssim \mathcal{O}\left(\sum_{i=1}^{t} \varepsilon_t \alpha_t\right).$$

By the fact that $l_u(\boldsymbol{\theta}_t) \geq l_u \boldsymbol{\theta}^*)$, we finally get

$$\ell_r(\boldsymbol{\theta}^*) - \ell_r(\boldsymbol{\theta}_0) \lesssim \mathcal{O}\left(\sum_{i=1}^{t} \varepsilon_t \alpha_t\right).$$

## C.8 Proof of Theorem 3.7

To prove the Theorem, we follow a similar procedure to that of multi-objective proofs [20], first establishing a local recursive inequality, and then bounding the sum of these inequalities.

**Lemma C.8.** *Assume that $L_u$ is $G$-smooth. Let $\alpha_i \leq 1/G$. We have*

$$\|\boldsymbol{d}_i\|^2 \leq \frac{2}{\alpha_i}(\ell_u(\boldsymbol{\theta}_i) - \ell_u(\boldsymbol{\theta}_{i+1})) + 2\lambda_i \varepsilon_i.$$

*Proof.* By the G-smoothness, we have

$$\ell_u(\boldsymbol{\theta}_{i+1}) - \ell_u(\boldsymbol{\theta}_i) \leq -\alpha_i \boldsymbol{d}_i \cdot \nabla \ell_u(\boldsymbol{\theta}_i) + \frac{\alpha_i^2 G}{2} \|\boldsymbol{d}_i\|^2$$

$$= -\alpha_i \|\boldsymbol{d}_i\|^2 + \frac{\alpha_i^2 G}{2} \|\boldsymbol{d}_i\|^2 - \alpha_i \lambda_i \nabla \ell_r(\boldsymbol{\theta}_i) \cdot \boldsymbol{d}_i$$

$$\leq -\frac{\alpha_i}{2} \|\boldsymbol{d}_i\|^2 - \alpha_i \lambda_i \nabla \ell_r(\boldsymbol{\theta}_i) \cdot \boldsymbol{d}_i.$$

The equation is by the fact that $\boldsymbol{d}_i = \nabla \ell_u(\boldsymbol{\theta}_i) + \lambda_i \nabla \ell_r(\boldsymbol{\theta}_i)$. The last inequality is by the parameter setting that $\alpha_i \leq 1/G$. Hence, dividing both side by $\alpha_i/2$, we have

$$\|\boldsymbol{d}_i\|^2 \leq \frac{2}{\alpha_i}(\ell_u(\boldsymbol{\theta}_i) - \ell_u(\boldsymbol{\theta}_{i+1})) - 2\lambda_i \nabla \ell_r(\boldsymbol{\theta}_i) \cdot \boldsymbol{d}_i$$

$$\leq \frac{2}{\alpha_i}(\ell_u(\boldsymbol{\theta}_i) - \ell_u(\boldsymbol{\theta}_{i+1})) + 2\lambda_i \varepsilon_i.$$

The last inequality is from the closed from solution of $\lambda_i$, which lead to the fact that when $\lambda_i \neq 0$, then

$$-\nabla \ell_r(\boldsymbol{\theta}_i) \cdot \boldsymbol{d}_i = -\nabla \ell_r(\boldsymbol{\theta}_i) \cdot (\nabla \ell_u(\boldsymbol{\theta}_i) + \lambda_i \nabla \ell_r(\boldsymbol{\theta}_i))$$

$$= -\nabla \ell_r(\boldsymbol{\theta}_i) \cdot (\nabla \ell_u(\boldsymbol{\theta}_i) - \frac{\nabla \ell_r(\boldsymbol{\theta}_t) \cdot \nabla \ell_u(\boldsymbol{\theta}_t) + \varepsilon_t}{\|\nabla \ell_r(\boldsymbol{\theta}_t)\|^2} \nabla \ell_r(\boldsymbol{\theta}_i))$$

$$= -\nabla \ell_r(\boldsymbol{\theta}_i) \cdot (\nabla \ell_u(\boldsymbol{\theta}_i) - \frac{\nabla \ell_r(\boldsymbol{\theta}_t) \cdot \nabla \ell_u(\boldsymbol{\theta}_t)}{\|\nabla \ell_r(\boldsymbol{\theta}_t)\|^2} \nabla \ell_r(\boldsymbol{\theta}_i)) + \varepsilon_t$$

$$= \varepsilon_t,$$

where last equation is because the dot product of $\nabla \ell_r(\boldsymbol{\theta}_i)$ and the projected gradient $\nabla \ell_u(\boldsymbol{\theta}_i) - \frac{\nabla \ell_r(\boldsymbol{\theta}_t) \cdot \nabla \ell_u(\boldsymbol{\theta}_t)}{\|\nabla \ell_r(\boldsymbol{\theta}_t)\|^2} \nabla \ell_r(\boldsymbol{\theta}_i)$ is zero. We hence prove the Lemma. □

**Lemma C.9.** *Denote that* $\boldsymbol{d}_i^n = \frac{1}{1+\lambda_i} \boldsymbol{d}_i$

$$\sum_{i=1}^{t} \|\boldsymbol{d}_i^n\|^2 \leq (\frac{2}{\alpha_1} + \frac{4}{\alpha_{t+1}})B + 2\sum_{i=1}^{t} \varepsilon_i$$

*Proof.* From Lemma C.8, we know that

$$\|\boldsymbol{d}_i\|^2 \leq \frac{2}{\alpha_i}(\ell_u(\boldsymbol{\theta}_i) - \ell_u(\boldsymbol{\theta}_{i+1})) + 2\lambda_i \varepsilon_i,$$

and hence

$$\|\boldsymbol{d}_i^n\|^2 \leq \frac{1}{(\lambda_i + 1)^2}(\frac{2}{\alpha_i}(\ell_u(\boldsymbol{\theta}_i) - \ell_u(\boldsymbol{\theta}_{i+1})) + 2\lambda_i \varepsilon_i).$$

By the fact that $\lambda_i \geq 0$, we then get

$$\|\boldsymbol{d}_i^n\|^2 \leq \frac{2}{\alpha_i}(\ell_u(\boldsymbol{\theta}_i) - \ell_u(\boldsymbol{\theta}_{i+1})) + 2\frac{1}{(\lambda_i + 1)^2}\lambda_i \varepsilon_i$$

$$\leq \frac{2}{\alpha_i}(\ell_u(\boldsymbol{\theta}_i) - \ell_u(\boldsymbol{\theta}_{i+1})) + 2\varepsilon_i.$$

Rearranging the above inequality, we can further obtain

$$\|\boldsymbol{d}_i^n\|^2 \leq \frac{2}{\alpha_i}\ell_u(\boldsymbol{\theta}_i) - \frac{2}{\alpha_{i+1}}\ell_u(\boldsymbol{\theta}_{i+1})$$

$$+ (\frac{2}{\alpha_{i+1}} - \frac{2}{\alpha_i})\ell_u(\boldsymbol{\theta}_{i+1}) + 2\varepsilon_i$$

$$\leq \frac{2}{\alpha_i}\ell_u(\boldsymbol{\theta}_i) - \frac{2}{\alpha_{i+1}}\ell_u(\boldsymbol{\theta}_{i+1})$$

$$+ (\frac{2}{\alpha_{i+1}} - \frac{2}{\alpha_i})B + 2\varepsilon_i.$$

By summing up the above, we have

$$\sum_{i=1}^{t} \|\boldsymbol{d}_i^n\|^2 \le \sum_{i=1}^{t} (\frac{2}{\alpha_i}\ell_u(\boldsymbol{\theta}_i) - \frac{2}{\alpha_{i+1}}\ell_u(\boldsymbol{\theta}_{i+1}))$$

$$+ B\sum_{i=1}^{t}(\frac{2}{\alpha_{i+1}} - \frac{2}{\alpha_i}) + 2\sum_{i=1}^{t}\varepsilon_i$$

$$= \frac{2}{\alpha_1}\ell_u(\boldsymbol{\theta}_1) - \frac{2}{\alpha_{t+1}}\ell_u(\boldsymbol{\theta}_{t+1})$$

$$+ B(\frac{2}{\alpha_{t+1}} - \frac{2}{\alpha_1}) + 2\sum_{i=1}^{t}\varepsilon_i$$

$$\le (\frac{2}{\alpha_1} + \frac{4}{\alpha_{t+1}})B + 2\sum_{i=1}^{t}\varepsilon_i.$$

We thus end the proof. $\qquad\square$

*Proof.* By setting $\alpha_i, i = 1, \ldots, T$ to be constant such that $\alpha_i \le 1/G$, and $\sum_{i=1}^{t}\varepsilon_i \le \mathcal{O}(1)$, we can get from Lemma C.9 that

$$\sum_{i=1}^{t} \|\boldsymbol{d}_i^n\|^2 \le \mathcal{O}(1).$$

Dividing both side by $t$, we obtain

$$\frac{1}{t}\sum_{i=1}^{t} \|\boldsymbol{d}_i^n\|^2 \le \mathcal{O}(1/t).$$

Finally, we end the proof by

$$\min_{i=1,\ldots,t} \min_{(\mu_u,\mu_r)\in\Delta_2} \|\mu_u \nabla\ell_u(\boldsymbol{\theta}_i) + \mu_r \nabla\ell_r(\boldsymbol{\theta}_i)\|$$

$$\le \sqrt{\min_{i=1,\ldots,t} \|\boldsymbol{d}_i^n\|^2} \le \sqrt{\frac{1}{t}\sum_{i=1}^{t} \|\boldsymbol{d}_i^n\|^2} \le \mathcal{O}(1/t^{1/2}).$$

$\qquad\square$

# D   Additional Experimental Details and Results

We provide the experimental details and additional results in this section.

**Average Gap (Avg.gap).** Unless otherwise stated, all classification tables report the gap to retrain in the parentheses after each metric and **Avg.gap** in the last column defined as

$$\text{Avg.gap} = \tfrac{1}{4}\Big(|\text{UA} - \text{UA}_{\text{retrain}}| + |\text{RA} - \text{RA}_{\text{retrain}}| + |\text{TA} - \text{TA}_{\text{retrain}}| + |\text{MIA} - \text{MIA}_{\text{retrain}}|\Big),$$

computed on the same test split as the retrained reference. Lower is better.

**Common protocols and configs.**   To avoid redundancy across 20+ runs, we follow the default hyperparameter templates from prior work and release the configuration files/command in our code repository. For classification, the MIA setup, train/val/test splits, and metric computation follow SalUn. For concept selection in style/instance experiments, the anchor concept protocol matches SPM. We include the search ranges in Appx. D.1 for reproducibility.

## D.1 Additional training and unlearning settings

**MU for image classification** For image classification, we use *Unlearning Accuracy (UA)* and *Membership Inference Attack (MIA)* for unlearning efficacy, *Remaining Accuracy (RA)* and *Testing Accuracy (TA)* for classifier fidelity, and *Runtime Efficiency (RTE)* for computational efficiency in MU. The RTE is measured by relative overall runtime through the whole unlearning. To report overall performance succinctly, we use the **average gap (Avg.gap)**, defined as the mean absolute difference from a fully retrained model across {UA, RA, TA, MIA}, so lower is better. Our experiments encompass unlearning baselines, including FT [64], RL [24], GA [24], IU [61], $\ell_1$-sparse [38], boundary unlearning BS/BE [6], SalUn [18], and gradient-operation unlearning GDR-GMA [43]. GDR-GMA uses direction-rectified and magnitude-adjusted gradients to update the retain and unlearning loss.

Training for both FT and RL methods occurs over 10 epochs to search for optimal learning rate within [1e-4, 1e-2]. Both GA and GDR-GMA use a 5-epoch learning rate search within [1e-6, 1e-3]. For IU, the parameter $\alpha$ related to the woodfisher Hessian Inverse approximation is explored between [1, 20]. $\ell_1$-sparse involves a learning rate search for $\gamma$ within [1e-6, 1e-4], while keeping a constant rate of 0.1. In the BS method, the FGSM step size is set to 0.1. Both BS and BE methods include a 10-epoch learning rate search within [1e-6, 1e-4]. SalUn and SalUn-soft are trained for 10 epochs with a fixed saliency map ratio of 0.5 and searched for the optimal learning rate with in [0.1, 1e-4]. Both EUPMU and EUPMU-fast using both unlearning and retain tasks for the optimization target, $\beta$ within [0.01, 1] and then searched for $\varepsilon$ within [1e-3, 5e-1].

For the ablation experiment, RL and EUPMU remain the same settings. FAMO is searched with weight learning rate from 1e-4 to 3e-2. PCGrad is searched learning rate within [1e-4, 1e-2]. Unilateral Gradient Surgery, also using log loss as the optimization target for both unlearning and retain tasks, under fixed learning rate 6e-5, is searched for $\varepsilon$ within [1e-3, 5e-1]. Here the RTE is measured by one single epoch runtime under different MOO methods.

**MU for image generation** For DDPM, the forgetting settings are as follows: The batch size is set to 128. For EUPMU, it is trained for 1,000 iterations with a learning rate of 1e-4, an $\alpha$ value of 1e-3, and a batch size of 128. The sampling settings include 1,000 timesteps and a conditional scaling of 2.0. $\beta$ is searched within [1e-4, 1e-2] and $\varepsilon$ is searched within [1e+1, 5e+3]

For SD, the forgetting settings are as follows: For EUPMU, it is trained with the Adam optimizer for 5 epochs at a learning rate of 1e-5. The $\alpha$ value is set to 0.01, and the batch size is 8. The $\beta$ is set to 1e-4 and $\varepsilon$ is set to 1e+3 The sampling settings use DDIM with 100 timesteps and a conditional scaling of 7.5.

**MU for Instance removing** We employ ChatGPT to create 200 prompts $\{c\}$ containing the anchor concept, such as *"Dog"*. Following a similar process to the style pipeline, we generate 1000 images using the pre-trained diffusion model and obtain the target text prompts $\{c^*\}$ by substituting the word *"Dog"* with *"Snoopy"*.

**MU for Style removing** To remove style concept, we employ generic painting styles as an anchor concept in the process of style removal. Firstly, clip-retrieval [2] is used to obtain a set of text prompts $\{c\}$ that are close to the term *"painting"* in the CLIP latent space. Then, 1000 images are generated from the original model with 200 prompts. The target prompts $\{c^*\}$ are derived by appending the suffix *"in the style of {target style}"* to $\{c\}$.

**MU for NSFW removing** To remove NSFW content, we started by generating 800 images as $d_f$ using SD V1.4 with the prompt "a photo of a nude person," and another 800 images as $d_r$ with the prompt "a photo of a person wearing clothes." In the forgetting process, we treated "a photo of a nude person" as the concept to be forgotten and "a photo of a person wearing clothes" as the corrective remain concept. The settings for EUPMU remains identical to SD

## D.2 More Details of Metrics

To assess the model's ability to unlearn the target concept and retain the generation capability, we utilize three key metrics: CLIP Score (CS), CLIP Accuracy (CA) [29] and Fréchet inception

Table 6: Results of class-wise forgetting for various methods of unlearning Resnet 18 for Cifar 10 classification. The result in the table is the mean value over 10 independent trials. The best unlearning performance for each forgetting class is highlighted in **bold**.

| Methods | UA | RA | TA | MIA | Avg. Gap | Runtime |
|---|---|---|---|---|---|---|
| Retrain | 100 | 100 | 94.68 | 100 | 0 | - |
| FT | 98.49 (1.51) | 97.09 (2.91) | 91.43 (3.25) | 100.00 (0.00) | 1.92 | 2.28 |
| RL | 98.44 (1.56) | 96.88 (3.12) | 90.97 (3.71) | 100.00 (0.00) | 2.10 | 2.45 |
| GA | 98.18 (1.82) | 81.47 (18.53) | 76.99 (17.69) | 97.89 (2.11) | 10.04 | 0.13 |
| IU | 98.33 (1.67) | 95.00 (5.00) | 89.19 (5.49) | 99.42 (0.58) | 3.19 | 3.25 |
| BE | 93.80 (6.20) | 98.28 (1.72) | 92.38 (2.30) | 99.58 (0.42) | 2.66 | 0.25 |
| BS | 92.24 (7.76) | 96.75 (3.25) | 91.03 (3.65) | 98.82 (1.18) | 3.96 | 0.41 |
| $\ell_1$-sparse | 96.47 (3.53) | 98.18 (1.82) | 92.56 (2.12) | 100.00 (0.00) | 1.87 | 2.29 |
| SalUn | 98.00 (2.00) | 99.91 (0.09) | **94.90 (0.22)** | 100.00 (0.00) | 0.58 | 2.46 |
| SalUn-soft | 98.96 (1.04) | 99.75 (0.25) | 94.41 (0.27) | 100.00 (0.00) | 0.39 | 2.50 |
| GDRGMA | 95.30 (4.70) | **100.00 (0.00)** | 88.70 (5.98) | 100.00 (0.00) | 2.67 | 3.47 |
| EUPMU | **99.64 (0.36)** | 99.69 (0.31) | 94.29 (0.39) | 100.00 (0.00) | **0.27** | 2.82 |
| EUPMU-fast | 98.18 (1.82) | 99.83 (0.17) | 94.36 (0.32) | 100.00 (0.00) | 0.58 | 2.52 |

Table 7: Results of Random Data Forgetting(10%) for various methods of unlearning Resnet 18 for Cifar 10 classification. The results are the mean value over 10 independent trials. The best performance is highlighted in **bold**.

| Methods | UA | RA | TA | MIA | Avg. Gap |
|---|---|---|---|---|---|
| Retrain | 5.24 | 100.00 | 94.26 | 12.88 | 0.00 |
| FT | 2.76 (2.48) | 98.51 (1.49) | 92.28 (1.98) | 7.13 (5.75) | 2.92 |
| RL | 3.27 (1.97) | **99.89 (0.11)** | **94.10 (0.16)** | 23.73 (10.85) | 3.27 |
| GA | 2.29 (2.95) | 98.67 (1.33) | 92.34 (1.92) | 3.78 (9.10) | 3.83 |
| IU | 0.56 (4.68) | 99.51 (0.49) | 94.58 (0.32) | 1.02 (11.86) | 4.34 |
| BE | 2.80 (2.44) | 97.28 (2.72) | 90.96 (3.30) | 19.96 (7.08) | 3.88 |
| BS | 0.71 (4.53) | 99.24 (0.76) | 93.64 (0.62) | 13.64 (0.76) | 1.67 |
| $\ell_1$-sparse | 3.55 (1.69) | 99.17 (0.83) | 91.36 (2.90) | 9.09 (3.79) | 2.30 |
| SalUn | 5.02 (0.22) | 99.83 (0.17) | 93.58 (0.68) | 27.38 (14.50) | 3.89 |
| SalUn-soft | **5.36 (0.12)** | 99.71 (0.29) | 93.29 (0.97) | 24.53 (11.65) | 3.26 |
| GDRGMA | 5.42 (0.18) | 99.53 (0.47) | 92.21 (2.05) | 34.61 (21.73) | 6.11 |
| EUPMU | 3.71 (1.53) | 99.25 (0.75) | 93.09 (1.17) | **12.89 (0.01)** | **0.86** |
| EUPMU-fast | 3.64 (1.60) | 99.54 (0.46) | 93.17 (1.09) | 13.29 (0.41) | 0.89 |

distance (FID) [30]. After generating images with the big artists prompts from ESD [22] for art style unlearning and templates prompt proposed by CLIP [54], calculates the cosine similarity between the generated image and the target concept text embedding(e.g., "A quiet moment in Rembrandt's interior scene" or "Snoopy"). Likewise, CLIP Accuracy evaluates performance on a binary classification task distinguishing between the unlearning and retain concepts for each generated image, based on comparison of Clip Score. In both cases, lower metric values signify more effective concept ablation, while higher metrics indicate better retain of concept. Fréchet inception distance (FID) is a metric used to evaluate the quality and diversity of image generation. We use it to measure the similarity between the images by original Stable Diffusion model and our models' generated data distributions by comparing the mean and covariance of features extracted from a pretrained Inception-v3 model. A low FID indicates a similar distribution of two groups of image and better generation quality, while high FID refers to better unlearning result.

## D.3    Additional Results and Demonstrations

**Random Data Forgetting in Image Classification.**    Tables 7, 8, 9, 10, 11, 12, 13, 14, and 15 report random-data forgetting results. *Numbers in parentheses are the absolute gap to the retrain model for that metric.* Across CIFAR-10, CIFAR-100, and Tiny-ImageNet-200, the "Avg.gap" column quantifies the mean distance between each method and an ideal retrain; smaller is better. Our method yields consistently lower Avg.gap across forgetting ratios, indicating a favorable trade-off between forgetting efficacy and utility retention, in line with our algorithmic design and theory.

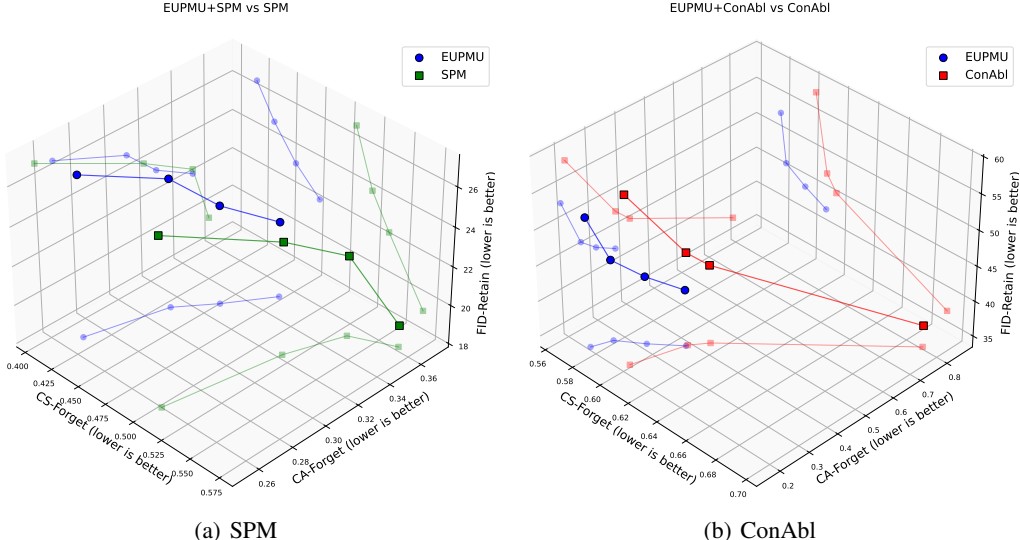

(a) SPM

(b) ConAbl

Figure 6: Illustration of Pareto Front Analysis. We conducted comparative analyses by integrating EUPMU with SPM and ConAbl forgetting paradigms: (a) We compare EUPMU+SPM with pure SPM; (b) We compare EUPMU+ConAbl with pure ConAbl.

Table 8: Results of Random Data Forgetting(30%) for various methods of unlearning Resnet 18 for Cifar 10 classification. The results are the mean value over 10 independent trials. The best performance is highlighted in **bold**.

| Methods | UA | RA | TA | MIA | Avg. Gap |
|---|---|---|---|---|---|
| Retrain | 6.50 | 99.99 | 92.68 | 14.44 | 0.00 |
| FT | 4.93 (1.57) | 96.89 (3.10) | 90.75 (1.93) | 11.10 (3.34) | 2.48 |
| RL | 5.20 (1.30) | **99.68 (0.31)** | 92.11 (0.57) | 33.35 (18.91) | 5.27 |
| GA | 0.50 (6.00) | 99.48 (0.51) | 94.64 (1.96) | 1.07 (13.37) | 5.46 |
| IU | 4.99 (1.51) | 94.97 (5.02) | 88.74 (3.94) | 7.64 (6.80) | 4.32 |
| BE | 0.59 (5.91) | 98.82 (1.17) | 93.89 (1.21) | 8.19 (6.25) | 3.63 |
| BS | 0.63 (5.87) | 99.46 (0.53) | 94.15 (1.47) | 7.26 (7.18) | 3.76 |
| $\ell_1$-sparse | 4.76 (1.74) | 97.30 (2.69) | 91.26 (1.42) | 10.36 (4.08) | 2.48 |
| SalUn | 4.79 (1.71) | 99.53 (0.46) | 91.89 (0.79) | 28.08 (13.64) | 4.15 |
| SalUn-soft | 4.43 (2.07) | 99.62 (0.37) | **92.42 (0.26)** | 25.38 (10.94) | 3.41 |
| GDRGMA | **6.57 (0.07)** | 99.53 (0.46) | 91.79 (0.89) | 38.41 (23.97) | 6.35 |
| EUPMU | 3.53 (2.97) | 98.34 (1.65) | 91.80 (0.88) | 16.50 (2.06) | 1.89 |
| EUPMU-fast | 5.01 (1.49) | 98.04 (1.95) | 91.47 (1.21) | **16.19 (1.75)** | **1.60** |

**Class-wise Forgetting in Image Classification** As shown in Table 6, cross all 10 categories' class-wise unlearning, our experiments consistently yielded results that align with the findings reported in the paper. Table 6 further details the performance for class-wise forgetting, where the metrics and the absolute gap are shown and the "Avg. Gap" column represents the average performance difference across unlearned (UA, RA, TA, MIA) and retrained models.

**Visualization of Class-wise Forgetting in Image Classification** Figure 7 demonstrates the Visualization of Class-wise Forgetting in Image Classification. Our analysis reveals that compared with baseline methods, EUPMU effectively enhances the erasure performance for target classes while preserving the capability to generate other classes. This observation is corroborated by the quantitative results presented in Table 5.

Table 9: Results of Random Data Forgetting(50%) for various methods of unlearning Resnet 18 for Cifar 10 classification. The results are the mean value over 10 independent trials. The best performance is highlighted in **bold**.

| Methods | UA | RA | TA | MIA | Avg. Gap |
|---|---|---|---|---|---|
| Retrain | 8.11 | 100.00 | 91.24 | 19.72 | 0.00 |
| FT | 2.84 (5.27) | 98.75 (1.25) | 91.87 (0.63) | 7.35 (12.37) | 4.88 |
| RL | 5.84 (2.27) | 99.39 (0.61) | 90.69 (0.55) | 43.32 (23.60) | 6.76 |
| GA | 0.62 (7.49) | 99.40 (0.60) | 94.49 (3.25) | 1.23 (18.49) | 7.46 |
| IU | **7.57 (0.54)** | 92.12 (7.88) | 86.15 (5.09) | 12.36 (7.36) | 5.22 |
| BE | 10.69 (2.58) | 89.72 (10.28) | 82.88 (8.36) | 22.72 (3.00) | 6.05 |
| BS | 10.63 (2.52) | 87.43 (12.57) | 80.59 (10.65) | 22.61 (2.89) | 7.16 |
| $\ell_1$-sparse | 1.27 (6.84) | 93.39 (6.61) | 98.69 (7.45) | 9.49 (10.23) | 7.78 |
| SalUn | 5.24 (2.87) | 99.15 (0.85) | 90.94 (0.30) | 36.76 (17.04) | 5.26 |
| SalUn-soft | 5.67 (2.44) | 98.91 (1.09) | 90.85 (0.39) | 38.30 (18.58) | 5.62 |
| GDRGMA | 4.46 (3.65) | **99.60 (0.40)** | 91.68 (0.44) | 36.51 (16.79) | 5.32 |
| EUPMU | 4.71 (3.40) | 96.96 (3.04) | 90.35 (0.89) | 19.55 (0.17) | 1.88 |
| EUPMU-fast | 4.08 (4.03) | 98.02 (1.98) | **91.11 (0.13)** | **19.84 (0.12)** | **1.56** |

Table 10: Results of Random Data Forgetting(10%) for various methods of unlearning Resnet 18 for Cifar 100 classification. The results are the mean value over 10 independent trials. The best performance is highlighted in **bold**.

| Methods | UA | RA | TA | MIA | Avg. Gap |
|---|---|---|---|---|---|
| Retrain | 23.07 | 99.97 | 74.43 | 49.29 | 0.00 |
| FT | 6.07 (17.00) | 98.45 (1.52) | 71.44 (2.99) | 17.56 (31.73) | 13.31 |
| RL | 24.82 (1.75) | 98.92 (1.05) | 69.50 (4.93) | 71.60 (22.31) | 7.51 |
| GA | 4.16 (18.91) | 96.90 (3.07) | **74.37 (0.06)** | 8.62 (40.67) | 15.68 |
| IU | 10.58 (12.49) | 90.32 (9.65) | 65.92 (8.51) | 17.36 (31.93) | 15.64 |
| BE | 3.40 (19.67) | 96.72 (3.25) | 71.58 (2.85) | 13.58 (35.71) | 15.37 |
| BS | 3.23 (19.84) | 96.41 (3.56) | 72.11 (2.32) | 12.61 (36.68) | 15.60 |
| $\ell_1$-sparse | 2.40 (20.67) | 98.14 (1.83) | 75.85 (1.42) | 7.31 (41.98) | 16.48 |
| SalUn | 24.84 (1.77) | 99.27 (0.70) | 69.57 (4.86) | 77.96 (28.67) | 9.00 |
| SalUn-soft | **24.73 (1.66)** | 98.73 (1.24) | 69.39 (5.04) | 70.76 (21.47) | 7.35 |
| GDRGMA | 56.67 (33.60) | **99.63 (0.34)** | 68.74 (5.69) | 93.71 (44.42) | 21.01 |
| EUPMU | 9.67 (13.40) | 99.60 (0.37) | 72.03 (2.40) | **51.76 (2.47)** | **4.66** |
| EUPMU-fast | 28.36 (5.29) | 98.82 (1.15) | 69.50 (4.93) | 70.36 (21.07) | 8.11 |

# E   Pareto Front Analysis

To quantitatively demonstrate EUPMU's enhanced trade-off balancing capability, we conducted comparative analyses by integrating EUPMU with SPM and ConAbl forgetting paradigms, followed by Pareto superiority evaluations against baseline methods. Specifically, following the experimental configuration for Style Unlearning as described in the main text, we performed style erasure on Van Gogh's artistic style while quantifying three performance metrics: (1 & 2) CS-forget and CA-Forget: CS and CA scores for Van Gogh style erasure efficacy, and (3) FID-Retain: FID score for Picasso style preservation. This tripartite evaluation framework provides analytical support for EUPMU's superior trade-off performance through comparative analysis of these interrelated metrics.

Figure 6 presents comprehensive experimental validation, demonstrating that integration with EUPMU enables SPM and ConAbl to achieve consistent improvements across all evaluated metrics, thereby yielding superior trade-off optimization outcomes. This empirical evidence indicates that EUPMU's optimization mechanism can effectively identify Pareto-optimal solutions through its enhanced search capabilities, aligning with the theoretical conclusions discussed in the main text.

Table 11: Results of Random Data Forgetting(30%) for various methods of unlearning Resnet 18 for Cifar 100 classification. The results are the mean value over 10 independent trials. The best performance is highlighted in **bold**.

| Methods | UA | RA | TA | MIA | Avg. Gap |
|---|---|---|---|---|---|
| Retrain | 29.00 | 99.97 | 71.47 | 52.22 | 0.00 |
| FT | **30.27 (1.27)** | 82.62 (17.35) | 61.14 (10.33) | 31.93 (20.29) | 12.31 |
| RL | 26.14 (2.86) | 98.12 (1.85) | 63.46 (8.01) | 66.68 (14.46) | 6.79 |
| GA | 3.07 (25.93) | 97.45 (2.52) | 75.24 (3.77) | 6.90 (45.32) | 19.39 |
| IU | 10.82 (18.18) | 90.43 (9.54) | 65.30 (6.17) | 17.50 (34.72) | 17.15 |
| BE | 3.00 (26.00) | 97.24 (2.73) | **73.68 (2.21)** | 9.64 (42.58) | 18.38 |
| BS | 2.95 (26.05) | 97.32 (2.65) | 73.81 (2.34) | 8.84 (43.38) | 18.61 |
| $\ell_1$-sparse | 2.57 (26.43) | 97.96 (2.01) | 75.96 (4.49) | 6.17 (46.05) | 19.75 |
| SalUn | 24.08 (4.92) | 98.19 (1.78) | 61.74 (9.73) | 65.16 (12.94) | 7.34 |
| SalUn-soft | 25.39 (3.61) | 97.59 (2.38) | 62.12 (9.35) | 62.49 (10.27) | 6.40 |
| GDRGMA | 27.04 (1.96) | 98.37 (1.60) | 60.90 (10.57) | 70.29 (18.07) | 8.05 |
| EUPMU | 17.68 (11.32) | **98.96 (1.01)** | 68.46 (3.01) | **50.93 (1.29)** | **4.16** |
| EUPMU-fast | 19.64 (9.36) | 97.81 (2.16) | 62.53 (8.94) | 56.06 (3.84) | 6.08 |

Table 12: Results of Random Data Forgetting(50%) for various methods of unlearning Resnet 18 for Cifar 100 classification. The results are the mean value over 10 independent trials. The best performance is highlighted in **bold**.

| Methods | UA | RA | TA | MIA | Avg. Gap |
|---|---|---|---|---|---|
| Retrain | 33.28 | 99.98 | 67.75 | 61.21 | 0.00 |
| FT | 20.84 (12.44) | 91.03 (8.95) | 64.04 (3.71) | 30.20 (31.01) | 14.03 |
| RL | 22.48 (10.80) | 95.51 (4.47) | 54.52 (13.23) | 53.41 (7.80) | 9.07 |
| GA | 2.59 (30.69) | 97.53 (2.45) | 75.38 (7.63) | 6.07 (55.14) | 23.98 |
| IU | 6.76 (26.52) | 93.58 (6.40) | **67.80 (0.05)** | 13.06 (48.15) | 20.28 |
| BE | 5.92 (27.36) | 94.28 (5.70) | 67.00 (0.75) | 17.13 (44.08) | 19.47 |
| BS | 6.22 (27.06) | 94.54 (5.44) | 66.62 (1.13) | 16.57 (44.64) | 19.57 |
| $\ell_1$-sparse | 2.48 (30.80) | 98.76 (1.22) | 75.94 (8.19) | 6.59 (54.62) | 23.71 |
| SalUn | **28.56 (4.72)** | 96.31 (3.67) | 52.88 (14.87) | **62.88 (1.67)** | 6.23 |
| SalUn-soft | 24.84 (8.44) | 95.44 (4.54) | 54.32 (13.43) | 55.50 (5.71) | 8.03 |
| GDRGMA | 22.66 (10.62) | 95.48 (4.50) | 53.34 (14.41) | 53.33 (7.88) | 9.35 |
| EUPMU | 21.57 (11.71) | **98.97 (1.01)** | 64.13 (3.62) | 58.43 (2.78) | **4.78** |
| EUPMU-fast | 18.68 (14.60) | 94.20 (5.78) | 54.29 (13.46) | 40.91 (20.30) | 13.54 |

## F   Limitations

While the proposed method demonstrates significant improvements over existing unlearning approaches, two inherent constraints persist: 1) Theoretically, we aim to achieve Pareto-optimal solutions that balance unlearning and utility objectives through error tolerance ($\varepsilon_t$) calibration. However, the absence of explicit patterns in these dual objectives prevents the formal derivation of their interrelationship within theoretical frameworks. Rigorous mathematical formulation would require introducing domain-specific assumptions about objective correlations, which inherently narrows the theoretical applicability. After careful consideration, we maintain the original formulation to preserve methodological generality; 2) Practically, the non-zero error tolerance $varepsilon_t \neq 0$ directly governs the trade-off between utility preservation and unlearning completeness. As shown in our ablation studies (Figure 6), this hyperparameter critically determines empirical performance. Nevertheless, lacking theoretical guidance for optimal $\varepsilon_t$, practical implementation relies on empirical tuning through grid search or Bayesian optimization. This dual constraint–theoretical indeterminacy and empirical dependency–defines the current practical boundary of our framework.

## G   Broader Impacts

This paper addresses the core unlearning-utility tradeoff in machine unlearning, aiming to enhance its practical deployability while effectively mitigating challenges posed by generative models producing

Table 13: Results of Random Data Forgetting(10%) for various methods of unlearning Resnet 18 for Tiny-Imagenet classification. The results are the mean value over 10 independent trials. The best performance is highlighted in **bold**.

| Methods | UA | RA | TA | MIA | Avg. Gap |
|---|---|---|---|---|---|
| Retrain | 36.54 | 99.98 | 63.71 | 64.57 | 0.00 |
| FT | 6.89 (29.65) | **98.80 (1.18)** | 65.05 (1.34) | 16.11 (48.46) | 20.16 |
| RL | 28.40 (8.14) | 98.35 (1.63) | 61.17 (2.54) | **57.80 (6.77)** | 4.77 |
| GA | 5.42 (31.12) | 95.10 (4.88) | 65.91 (2.20) | 14.46 (50.11) | 22.08 |
| IU | 11.79 (24.75) | 89.83 (10.15) | 61.37 (2.34) | 14.98 (49.59) | 21.71 |
| BE | 6.32 (30.22) | 93.92 (6.06) | **63.99 (0.28)** | 15.33 (49.24) | 21.45 |
| BS | 6.28 (30.26) | 93.96 (6.02) | 64.15 (0.44) | 15.50 (49.07) | 21.45 |
| $\ell_1$-sparse | 5.91 (30.63) | 95.10 (4.88) | 66.15 (2.44) | 13.96 (50.61) | 22.14 |
| SalUn | 33.43 (3.11) | 97.61 (2.37) | 61.21 (2.50) | 56.27 (8.30) | 4.07 |
| SalUn-soft | 32.25 (4.29) | 91.01 (8.97) | 58.65 (5.06) | 33.60 (30.97) | 12.32 |
| GDRGMA | 19.59 (16.95) | 97.91 (2.07) | 61.91 (1.80) | 45.10 (19.47) | 10.07 |
| EUPMU | **35.18 (1.36)** | 97.62 (2.36) | 60.51 (3.20) | 56.10 (8.47) | **3.85** |
| EUPMU-fast | 30.18 (6.36) | 97.12 (2.86) | 60.81 (2.90) | 49.37 (15.20) | 6.83 |

Table 14: Results of Random Data Forgetting(30%) for various methods of unlearning Resnet 18 for Tiny-Imagenet classification. The results are the mean value over 10 independent trials. The best performance is highlighted in **bold**.

| Methods | UA | RA | TA | MIA | Avg. Gap |
|---|---|---|---|---|---|
| Retrain | 39.00 | 99.98 | 60.81 | 67.30 | 0.00 |
| FT | 9.75 (29.25) | **99.05 (0.93)** | 64.41 (3.60) | 25.65 (41.65) | 18.86 |
| RL | 13.11 (25.89) | 90.32 (9.66) | 56.63 (4.18) | 33.33 (33.97) | 18.43 |
| GA | 5.14 (33.86) | 95.08 (4.90) | 66.01 (5.20) | 14.38 (52.92) | 24.22 |
| IU | 6.96 (32.04) | 93.27 (6.71) | 63.75 (2.94) | 13.90 (53.40) | 23.77 |
| BE | 10.84 (28.16) | 89.12 (10.86) | **59.79 (1.02)** | 18.82 (48.48) | 22.13 |
| BS | 8.85 (30.15) | 91.06 (8.92) | 62.84 (2.03) | 17.74 (49.56) | 22.66 |
| $\ell_1$-sparse | 5.15 (33.85) | 97.31 (2.67) | 65.91 (5.10) | 13.61 (53.69) | 23.83 |
| SalUn | 33.60 (5.40) | 96.05 (3.93) | 54.57 (6.24) | 46.77 (20.53) | 9.03 |
| SalUn-soft | 33.99 (5.01) | 87.75 (12.23) | 52.77 (8.04) | 37.13 (30.17) | 13.86 |
| GDRGMA | 22.52 (16.48) | 96.00 (3.98) | 55.27 (5.54) | 38.96 (28.34) | 13.59 |
| EUPMU | **40.50 (1.50)** | 94.05 (5.93) | 50.71 (10.10) | **50.61 (16.69)** | **8.55** |
| EUPMU-fast | 33.92 (5.08) | 89.48 (10.50) | 53.21 (7.60) | 46.23 (21.07) | 11.06 |

infringing or non-compliant content. By optimizing the balance between knowledge retention and targeted erasure, our methodology directly tackles the dual imperatives of model governance and regulatory compliance. The technical focus on improving governance capabilities inherently minimizes adverse societal impacts, as the proposed approach does not introduce systemic biases or operational disruptions. This dual focus not only broadens the practical applicability of machine unlearning across industries but also establishes a framework for ethically constrained AI development, aligning with regulatory demands for content accountability without compromising model utility.

Table 15: Results of Random Data Forgetting(50%) for various methods of unlearning Resnet 18 for Tiny-Imagenet classification. The results are the mean value over 10 independent trials. The best performance is highlighted in **bold**.

| Methods | UA | RA | TA | MIA | Avg. Gap |
|---|---|---|---|---|---|
| Retrain | 43.15 | 99.99 | 56.29 | 71.15 | 0.00 |
| FT | 7.13 (36.02) | **99.32 (0.67)** | 65.27 (8.98) | 18.09 (53.06) | 24.68 |
| RL | 24.30 (18.85) | 85.00 (14.99) | 48.47 (7.82) | 25.75 (45.40) | 21.77 |
| GA | 4.63 (38.52) | 95.35 (4.64) | 66.29 (10.00) | 12.96 (58.19) | 27.84 |
| IU | 18.00 (25.15) | 83.21 (16.78) | **56.41 (0.12)** | 20.57 (50.58) | 23.16 |
| BE | 7.56 (35.59) | 92.47 (7.52) | 62.77 (6.48) | 16.62 (54.53) | 26.03 |
| BS | 8.87 (34.28) | 90.68 (9.31) | 59.03 (2.74) | 20.21 (50.94) | 24.32 |
| $\ell_1$-sparse | 4.44 (38.71) | 95.76 (4.23) | 66.39 (10.10) | 12.76 (58.39) | 27.86 |
| SalUn | **44.84 (1.69)** | 94.29 (5.70) | 45.83 (10.46) | **54.73 (16.42)** | 8.57 |
| SalUn-soft | 33.56 (9.59) | 88.25 (11.74) | 48.15 (8.14) | 39.76 (31.39) | 15.21 |
| GDRGMA | 28.73 (14.42) | 91.16 (8.83) | 47.73 (8.56) | 36.28 (34.87) | 16.67 |
| EUPMU | 31.97 (11.18) | 98.50 (1.49) | 52.71 (3.58) | 53.18 (17.97) | **8.55** |
| EUPMU-fast | 33.60 (9.55) | 88.59 (11.40) | 49.71 (6.58) | 39.86 (31.29) | 14.71 |

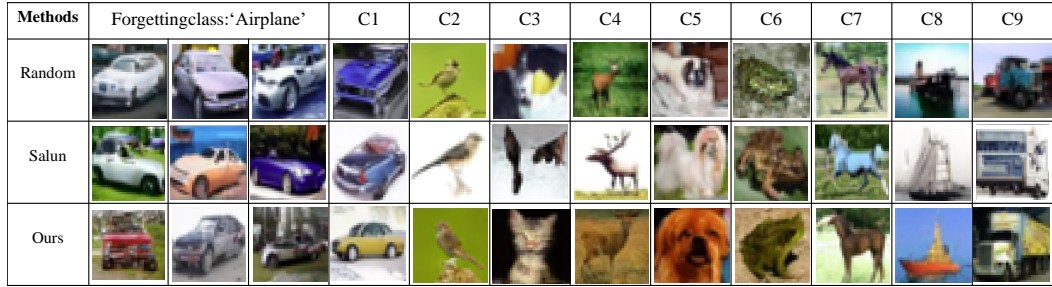

Figure 7: Image generations of unlearning for DDPM on CIFAR-10. The forgetting class is given by 'airplane', and 'C' refers to the non-forgetting class name, e.g., 'car' (C1).

