# OpenReview forum: "Efficient Utility-Preserving Machine Unlearning with Implicit Gradient Surgery"
_NeurIPS.cc/2025/Conference — NeurIPS 2025 poster_

### Official Review · Reviewer_7Sf9 · 2025-06-25

**Clarity:** 2
**Significance:** 3
**Originality:** 3
**Rating:** 4
**Confidence:** 3

**Summary:**

Machine unlearning aims to remove the memory of specific data from the pre-trained model, and there is a trade-off between increasing the effectiveness of unlearning and maintaining the performance of the original model. One way to mitigate this is to employ a multi-objective optimization, but existing methods are inadequate for unlearning. To address this issue, the paper formulates a constrained optimization problem that aims to optimize the unlearning objective under the constraint of a bounded increase for utility loss and proposes an efficient method based on gradient surgery to solve this problem.

**Questions:**

1. Could you clarify the precise definition of "pure linearization" used in Table 5 and provide a citation or detailed explanation of "RL" in Section 4.3?
2. How sensitive is the performance to the value of the error tolerance $\varepsilon$?
3. Have you analyzed why the proposed method becomes less cost-effective specifically in the random forgetting scenario?
4. Why is the evaluation process different from existing studies?

**Ethical Concerns:**

["NO or VERY MINOR ethics concerns only"]

**Final Justification:**

The authors’ rebuttal addresses most of my concerns, and I have accordingly increased my initial score.

**Limitations:**

The authors discussed the limitations in the appendix.

**Paper Formatting Concerns:**

There are no major formatting concerns in the paper.

**Quality:**

3

**Strengths And Weaknesses:**

## Strengths
1. This paper accurately provides why multi-objective optimization cannot be directly applied to the challenges of machine unlearning.
2. Each proposition is thoroughly proven in the appendix.
3. The paper presents comprehensive experimental results, and the image generation outcomes clearly illustrate the effectiveness of the proposed method.

## Weaknesses
1. While existing unlearning studies [1, 2] are evaluated on datasets with diverse classes such as CIFAR-100, Tiny-ImageNet, and ImageNet-1k, the paper only conducts experiments on relatively simple datasets with 10 classes, such as CIFAR-10 and Imagenette. As a result, the evaluation appears insufficient to fully validate the generalizability and scalability of the proposed method.
2. In Section 4.3, the definition of "pure linearization" is missing. There is no citation or clear description for "RL". Table 5 shows the proposed method takes more computation time than "pure linearization". If "pure linearization" refers to "linear weighting," then it raises concerns about the effectiveness of the proposed method, which claims to achieve comparable computational efficiency to "linear weighting".
3. While it is understandable that the hyperparameter "error tolerance" $\varepsilon$ influences empirical performance, there is no sensitivity analysis of the performance to $\varepsilon$. This lack of analysis makes it difficult to assess the robustness of the proposed method.
4. Table 6 shows that the proposed method incurs a significantly higher computational cost compared to GA in the random data forgetting scenario, while achieving only marginal performance gains. This raises questions about its cost-effectiveness.
5. While existing methods [1, 2] evaluate unlearning performance by quantifying the gap between an unlearned model and a model retrained from scratch without data subject to unlearning, this paper evaluates performance using raw accuracy values. Given that the objective of unlearning is to obtain a model equivalent to one trained without using the data, it would be more appropriate to evaluate performance based on the gap from the retrained model trained only on the retained data.

Minor:

6. Some highlights in Tables 5, 6, and 7 are incorrect.
7. There is a typo in line 761 of Appendix F: "$varepsilon_t$" should be "$\varepsilon_t$".

References

[1] Fan et al., Salun: Empowering machine unlearning via gradient-based weight saliency in both image classification and generation, In Proc. ICLR, 2024.

[2] Jia et al., model sparsity can simplify machine unlearning, In Proc. NeurIPS, 2023.

---

> ### Author Rebuttal · Authors · 2025-07-31
>
> We would like to thank reviewer 7Sf9 for the insightful comments. We address the concerns below.
>
> **W1. Datasets are relatively simple...the evaluation appears insufficient to fully validate the generalizability and scalability of the proposed method**
>
> Good suggestion! Our study deliberately spans multiple data regimes—natural-image classification, generative diffusion, style/instance manipulation, and safety filtering. We also evaluate on different backbones of Resnet, VIT, DDPM and Stable Diffusion. Each results are averaged over multiple independent trials in more than one different code base implementation. Therefore, we believe that such diverse experimental settings could already be able to prove the generalizability and scalability of the proposed method.
>
> Meanwhile, we tested the performance of Resnet 18 unlearning on Cifar 100, random data forgetting 10%.
>
> | **Methods** | **UA** | **RA** | **TA** | **MIA** | **Avg. Gap 4** | **Avg. Gap 2** |
> | :--- | :--- | :--- | :--- | :--- | :--- | :--- |
> | Retain | 26.47 | 99.97 | 74.13 | 51.00 | 0.00 | 0.00 |
> | FT | 2.42 (24.05) | 99.95 (**0.02**) | 75.55 (1.42) | 11.04 (39.96) | 16.36 | 20.69 |
> | RL| 55.03 (28.56) | 99.81 (0.16) | 70.03 (4.09) | 98.97 (47.97) | 20.20 | 26.03 |
> | GA | 3.13 (23.34) | 97.33 (2.64) | 75.31 (1.18) | 7.24 (43.76) | 17.73 | 22.47 |
> | IU | 3.18 (23.29) | 97.15 (2.82) | 73.49 (0.64) | 9.62 (41.38) | 17.03 | 21.01 |
> | BE | 2.31 (24.16) | 97.27 (2.70) | 73.93 (**0.20**) | 9.62 (41.38) | 17.11 | 20.79 |
> | BS | 2.27 (24.20) | 97.41 (2.56) | 75.26 (1.13) | 5.82 (45.18) | 18.27 | 23.16 |
> | l₁-Sparse | 10.64 (15.83) | 96.62 (3.35) | 70.99 (3.14) | 22.58 (28.42) | 12.68 | 15.78 |
> | SalUn | 27.53 (**1.06**) | 97.00 (2.97) | 67.79 (6.34) | 70.79 (19.79) | 7.54 | 13.07 |
> | SalUn-soft| 24.24 (2.23) | 98.95 (1.02) | 70.48 (3.65) | 79.13 (28.13) | 8.76 | 15.89 |
> | **EUPMU(ours)** | 13.53 (12.94) | 97.02 (2.95) | 71.90 (2.23) | 51.02 (**0.02**) | **4.53** | **1.12** |
>
>
> These results are consistent with the conclusions presented in the paper. We will include results on larger-scale datasets, such as Tiny-ImageNet and ImageNet-1k, in a future version of the paper.
>
>
> **W2 \& Q1.Define "pure linearization"  and "RL"...it raises concerns about the effectiveness of the proposed method, which claims to achieve comparable computational efficiency to "linear weighting**
>
> Thank you for pointing this out! "RL" refers to Random-Labeling as cited in Appendix D.1, and pure linearization indeed refers to "linear weighting".
>
> The introduction of an approximation to the gradient method for the Lagrangian (equation 4) avoids the need to explicitly solve for the two objectives. However, the computation of $\ell_r (\theta_{t+1})$ (equation 8) introduces an additional forward pass. This results in a slight increase in computational complexity compared to linearization, although the computational cost is still significantly lower than that of standard backpropagation. This explains the slight increase in computational cost shown in Table 5.
>
> However, this additional forward pass can be eliminated by reusing the value of $\ell_r (\theta_{t+1})$ computed during the forward pass of the next iteration (*t*+1). While this may lead to a mismatch between the data batches at step *t* and *t*+1 during stochastic training, it does not affect the model's convergence due to the unbiased nature of the stochastic process. Our latest experiments show that this modified approach (see EUPMU (modified)) achieves similar results comparable to EUPMU (previous) in Table 5, with a computational time on par with linearization.
>
>
> | Methods | UA | RA | TA | MIA | Avg. score | RTE |
> | :---: | :---: | :---: | :---: | :---: | :---: | :---: |
> | Linearization  | 14.11  | 90.92 | 86.05  | 20.16  | 14.325 | **15.7** |
> | **EUPMU (previous)**  | **0.49**  | **99.88** | 94.42  | **1.75**  | **1.985** | 18.4 |
> | **EUPMU (modified)**  | 0.51  | 99.79 | **94.45**  | 1.79  | 2.015 | **15.7** |
>
> We will discuss this more lightweight computational approach and provide sufficient experimental validation in a future version of the paper.
>
> **W3 \& Q2. Error tolerance $\varepsilon$ sensitivity...This lack of analysis makes it difficult to assess the robustness of the proposed method**
>
> Below, we provide an expanded ablation study error tolerance $\varepsilon$.
>
> | $\varepsilon$ | Retain Loss | Forget Loss |
> | :---: | :---: | :---: |
> | 0.00  | 0.052  |   **2.25** |
> | 0.10 | 0.035  | 2.53  |
> | 0.25 | 0.031  | 2.92  |
> | 0.50 | 0.029  | 3.58  |
> | 1.00 | **0.028** | 4.41  |
>
> We observe that as $\varepsilon$ ranges from 0 to 1, our algorithm smoothly traces the Pareto front. This confirms that the utility-unlearning tradeoff is controllably adjustable, allowing practitioners to select a balance suitable for their specific needs. This finding is in direct alignment with our statements in Remark 3.3 and Remark 3.6.
>
> Regarding the tuning of $\varepsilon_t$, this process is not burdensome. As guided by our theoretical results in Remark 3.3 and Theorem 3.7, one can effectively narrow the search range based on the magnitude of the loss function, which minimizes practical tuning effort. We will add further experiments and discussion to clarify this in the revised manuscript.
>
> **W4 \& Q3. Cost‑effectiveness vs. GA in Table 6**
>
> The running time comparison in Table 6 requires careful context. Fundamentally, the per-iteration computational cost of EUPMU (with the optimizations from our W2 response) and Gradient Ascent (GA) is the same. The lower time for GA is an artifact of its training instability: it must be early-stopped at 20% of epochs to prevent gradient explosion, which would otherwise cause a catastrophic drop in Test Accuracy (TA).
>
> If run for an equal number of epochs, EUPMU would have a similar runtime to GA but achieve a higher TA due to better utility preservation.
>
> Moreover, GA's result in Table 6 is a special case. It performs poorly on all other datasets (e.g., Table 7). This highlights the advantage of our method, which provides a superior trade-off, delivering more effective unlearning and utility preservation without a true increase in computational cost.
>
> **W5 \& Q4. Evaluate gap‑to‑retrain metric.**
>
> From a practical standpoint, directly optimizing for raw accuracy is more beneficial for real-world applications. In practice, the results of a fully retrained model (the "retrain oracle") are often unavailable. As long as the model's utility and data privacy metrics are guaranteed, its practical effectiveness is ensured. In our experiments, the unlearning process further fine-tunes the model on the retained data via the retain_loss. This can lead to performance that is better than retraining from scratch, especially on simpler tasks (like Table 6). In such cases, a comparison based on the "gap-to-retrain" metric would not be fair. Therefore, we chose to present our results using raw accuracy.
>
> If the goal is to minimize the gap to the Retrain Oracle, EUPMU still outperforms other methods. We benchmarked against GDR-GMA [1] and Learn to Unlearn [2]—strong baselines noted by reviewers and established as state-of-the-art in their publications. The following table presents the results for class-wise unlearning with ResNet-18 on CIFAR-10:
>
> | Method | Retain Acc | Forget Acc | Test Acc | MIA  | Avg Gap  |
> | --- | --- | --- | --- | --- | --- |
> | Retrain Oracle | 99.42 % (0.00) | 0.00 % (0.00) | 94.08 % (0.00) | 0.000 (0.000) |   0.00 |
> | GDR-GMA  | 95.30 % (4.12) | 0.00 % (**0.00**) | 88.70 % (5.38) | 0.000 (0.000) |   2.38 |
> | Learn to Unlearn | 98.98 % (**0.44**) | 0.76 % (0.76) | 92.00 % (2.08) | 0.000 (0.000) |   0.82 |
> | **EUPMU (Ours)** | 98.75 % (0.67) | 0.09 % (0.09) | 92.18 % (**1.90**) | 0.000 (0.000) |   **0.67** |
>
> As shown in the above gap‑to‑retrain results (and also the table in the response of W1), when we optimize the hyperparameters for gap-to-retrain, EUPMU still performed the best compared to other methods as well.
>
> [1]Lin, Shen, et al. "GDR-GMA: Machine Unlearning via Direction-Rectified and Magnitude-Adjusted Gradients." Proceedings of the 32nd ACM International Conference on Multimedia. 2024.
>
> [2] Qu, Youyang, et al. "Learn to Unlearn: A Survey on Machine Unlearning." *arXiv preprint arXiv:2305.07512* (2023).

---

> ### Author Response · Authors · 2025-08-02
> **Further Clarification on Cost-Effectiveness (W2 & W4)**
>
> Dear Reviewer 7Sf9,
>
> Thank you again for your valuable feedback. To further clarify our method's cost-effectiveness (W2 & W4) and the practical trade-offs involved, we have conducted a targeted experiment on the random data forgetting task, using the gap-to-retrain metric as the key benchmark. We also have implemented and tested the modified EUPMU with improved efficiency mentioned in response to W2, using next step's retain loss to measure the change rather than doing an extra forward pass on the current batch.
>
> The results below are presented to directly address your concerns, and they lead to two clear, decisive conclusions:
>
> | Method | Unlearning Accuracy (%) | Retain Accuracy (%) | Test Accuracy (%) | MIA | Average Gap | RTE |
> | :--- | :--- | :--- | :--- | :--- | :--- | :--- |
> | Retrain | 5.24 | 100.0 | 94.26 | 12.88 | 0.00 | 43.29 |
> | GA | 0.69(4.55) | 99.50(0.50) | **94.01(0.25)** | 1.70(11.18) | 4.12 | 0.13 |
> | RL | 7.61(2.37) | **99.67(0.33)** | 92.83(1.43) | 37.36(24.48) | 7.15 | 2.64 |
> | GDR-GMA | 0.51(4.73) | 99.55(0.45) | **94.51(0.25)** | 0.98(11.90) | 4.33 | 3.47 |
> | Learn to Unlearn | 2.69(2.55) | 97.73(2.27) | 89.78(4.48) | 9.09(3.79) | 3.27 | 4.63 |
> | EUPMU_RL | 3.64(1.60) | 99.54(0.46) | 93.17(1.09) | 13.29(0.41) | **0.89** | 2.69 |
> | EUPMU_GA | **5.58(0.34)** | 95.46(4.54) | 89.02(5.24) | 11.20(1.68) | 2.95 | 1.31 |
> | EUPMU_GA (1 epoch) | 6.42(1.18) | 94.17(5.83) | 88.43(5.83) | **12.96(0.08)** | 3.23 | 0.19 |
>
> 1.  Our Method is a *Flexible Highly Effective and Lightweight Framework*. Our approach is best understood as a "plug-and-play" framework that enhances existing unlearning methods. Crucially, this integration comes at **no significant computational cost**. As shown in the table, the runtime (RTE) of EUPMU RL (2.69) is nearly identical to the baseline RL (2.64), yet it dramatically reduces the Average Gap from 7.15 to an exceptional **0.89**.
>
> 2.  A Direct Rebuttal to GA's Apparent Cost-Effectiveness. In our initial rebuttal, we argued that GA's low runtime is a result of its training instability, which requires aggressive early stopping. This experiment provides the definitive evidence. When our EUPMU_GA is also run for only **one epoch**—matching the conditions under which GA is viable—it achieves a much better Average Gap of **3.23** compared to GA's 4.12, at a comparable runtime (RTE 0.19 vs 0.13). This proves our method's performance gains stem not from longer training, but from a fundamentally more stable and effective optimization process per update step.
>
> We believe this analysis confirms that our framework provides a superior trade-off between unlearning effectiveness, model utility, and computational cost. We are confident this resolves the questions regarding practicality and will incorporate this analysis into the final manuscript.

---

> ### Comment · Reviewer_7Sf9 · 2025-08-04
>
> I appreciate the authors' thoughtful responses and the additional experimental results provided.
> I understand that the proposed method can be applied across a diverse set of tasks, including image classification, generative diffusion, style/instance manipulation, and safety filtering.
>
> However, I remain concerned about the response to W5 and Q4.
> The goal of machine unlearning is to completely remove the influence of specific training data such that the resulting model behaves equivalently to one that was trained without those data [1]. From this perspective, I believe that the most appropriate and principled evaluation metric is the Gap, i.e., the difference in performance between the unlearned model and the retrained model trained from scratch without the forgotten data.
> In this regard, I observe that in Table 6 of the submission and in the response to W1, the proposed method (UA) performs worse than the baseline in terms of Gap, which raises concerns about the effectiveness of the proposed method.
> Moreover, in the new results presented in the response to W5 and Q4, it is shown that UA achieves a smaller Gap than the baselines. However, I am unclear whether this improvement is the result of tuning the method based on the retrained model's accuracy. If so, this would be problematic, as such tuning undermines the goal of achieving effective unlearning without access to retraining.
> I would appreciate clarification on this point.
>
>
> [1]Xu et al., Machine Unlearning: A Survey. ACM Computing Surveys, 2023.

---

> > ### Author Response · Authors · 2025-08-05
> > **Response to Reviewer 7Sf9 (follow‑up to W5 & Q4)**
> >
> > We thank you for the clear articulation of your remaining concern.
> >
> > ### 1. *“I believe that the most appropriate and principled evaluation metric is the **Gap** …”*
> >
> > Insightful question! We agree that, in an ideal unlearning system, the **Gap‑to‑Retrain** should be 0. In the experiments, we can categorize the unlearning tasks as follows.
> >
> > * **Typical tasks (class, person, style, safety unlearning).**  Here the forgotten data are *semantically distinct* from the retained data. The retrain‑oracle therefore achieves the best utility and privacy simultaneously, so **optimizing absolute validation metrics automatically minimises Gap**. For example, forgets an image class in CIFAR-10 with ResNet-18 classification on Cifar 10 (table in the response to W5 & Q4), EUPMU reaches the best average score for overall and gap-to-retrain performance;
> >
> > * **Atypical task (random‑forget).** Forgotten and retained samples are i.i.d.; the retrain‑oracle may be *not* guaranteed to be best on every metric, so absolute performance and Gap can diverge. This benchmark is widely used for stress‑testing but is far less common in practice (e.g., rights‑to‑be‑forgotten requests normally target a specific user or class).
> >
> > ### 2. *“I observe that in Table 6 of the submission and in the response to W1, the proposed method performs worse than the baseline in terms of Gap, which raises concerns …”*
> >
> > We would like to make the following clarification.
> >
> > *   **For Table 6 (CIFAR-10 random forget):** The experiment in Table 6 involved random forgetting, which, as classified in point 1, is an atypical task. Given the special nature of this benchmark, the results we originally presented in Table 6 were optimised for absolute metrics. In the table provided in the Further Clarification on Cost-Effectiveness (W2 & W4), we show that with a different choice of hyperparameters, our method can also achieve a significantly better average gap on the same task.
> >
> > *   **For the table in the response to W1 (CIFAR-100 random forget):** We apologise for not labeling the table in response to W1 more clearly. This may have led to the misunderstanding that our method underperforms the baseline. The two aggregated numbers are:
> >
> > | Column | Definition | Meaning |
> > | :--- | :--- | :--- |
> > | **Avg. Gap 4** | Average gap on *all four* metrics (Retain‑Acc, Forget‑Acc, Test‑Acc, MIA) | Overall performance |
> > | **Avg. Gap 2** | Average gap on the *two test‑time* metrics (Test‑Acc, MIA) | Generalization performance |
> >
> > As shown in the table, Avg. Gap 4 reflects the overall average performance, while Avg. Gap 2 reflects generalization performance. The performance on the test set is more indicative of an algorithm's practical performance. We observe that our method already had the best Avg. Gap 4 (4.53 vs 7.54), and an even larger margin on Avg. Gap 2 (1.12), comparing to other methods. This sufficiently demonstrates the effectiveness of our method. Furthermore, our larger advantage on the two test-time metrics highlights its strong generalization capability.
> >
> > ### 3. *“In the new results (W5 & Q4) UA now has a smaller Gap; was this achieved by tuning on the retrained model?”*
> >
> > In fact, for every method, at no stage did we use retrain‑oracle metrics for hyper‑parameter selection; all baselines received identical treatment. We adjust the hyperparameters based on validation performance.
> >
> > As mentioned in point 1, the tasks in our experiments for W5 & Q4 are typical tasks. In this scenario, the forget and retain sets are semantically distinct, and the retrain-oracle achieves the best performance across all metrics. Therefore, optimizing for validation performance is equivalent to optimizing for the Gap-to-Retrain, which is why our method achieved the best results.
> >
> > We will make these points explicit in the revision and add a paragraph clarifying why random‑forget behaves differently. Your feedback has materially improved the paper—thank you again for the close reading and constructive guidance.

---

> > > ### Comment · Reviewer_7Sf9 · 2025-08-06
> > >
> > > I would like to thank the authors once again for their detailed and thoughtful responses.
> > > The authors’ rebuttal has addressed most of my concerns, and I have decided to raise my score.
> > > Regarding the 3rd point, I had misunderstood the tasks in the experiments for W5 and Q4. I apologize for the confusion and any inconvenience this may have caused.
> > >
> > > In addition, considering the objectives of machine unlearning, I think it is important to evaluate by comparing with a retrained model that does not retain information of the specific forgetting data in the atypical task.
> > > I believe that incorporating the discussions so far, as well as the new experimental results, into the revised version will further improve the paper.

---

> > > > ### Author Response · Authors · 2025-08-06
> > > >
> > > > Dear Reviewer 7Sf9,
> > > >
> > > > Thank you very much for your comment, for acknowledging our rebuttal, and for raising your score. We are delighted that we were able to address most of your concerns.
> > > >
> > > > The discussion with you has been incredibly insightful and has given us a much clearer perspective on how to improve the paper's presentation, especially regarding the nuances of evaluation.
> > > >
> > > > We fully agree with your final point on the importance of the gap-to-retrain metric, particularly for atypical tasks. As you suggested, we will be sure to integrate all of our new results and the detailed discussion on evaluation methodologies into the revised manuscript.
> > > >
> > > > Thank you once again for your constructive engagement, which has helped us to significantly strengthen our work.
> > > >
> > > > Best regards,
> > > >
> > > > The Authors

---

### Official Review · Reviewer_mcr2 · 2025-07-03

**Clarity:** 3
**Significance:** 3
**Originality:** 3
**Rating:** 4
**Confidence:** 4

**Summary:**

This paper addresses the challenge of balancing unlearning efficacy and utility preservation in machine unlearning (MU). The authors formulate the Utility Preserving Unlearning Problem as a constrained optimization: maximize the decrease in the unlearning loss under a bounded increase in the retaining loss. Empirical evaluations on image classification and generation tasks demonstrate that EUPMU outperforms baselines.

**Questions:**

Please see the weaknesses.

**Ethical Concerns:**

["NO or VERY MINOR ethics concerns only"]

**Final Justification:**

The authors' responses have addressed my concerns, and thus I decide to maintain my score.

**Quality:**

3

**Strengths And Weaknesses:**

Strength:

1. It is insightful that formulating MU as a constrained optimization problem with explicit control over utility loss.
2. This paper provides detailed theoretical analysis and experimental evaluations that show consistent performance improvements.

Weakness:

1. Although it is very interesting to formulate MU as a gradient surgery problem, the gradient projection methods on MU have been investigated in the existing works [1][2]. I suggest that the authors add the corresponding discussions and comparison experiments.
2. Some important methods are missed on the SD models, e.g., MACE [3] and UCE [4].
3. The ablation experiments on the hyperparameters should also be added.
4. Is it possible to implement EUPMU for multi-concept unlearning on the SD models?

[1] Lin S, Zhang X, Susilo W, et al. GDR-GMA: Machine Unlearning via Direction-Rectified and Magnitude-Adjusted Gradients[C]//Proceedings of the 32nd ACM International Conference on Multimedia. 2024: 9087-9095.

[2] Hoang T, Rana S, Gupta S, et al. Learn to unlearn for deep neural networks: Minimizing unlearning interference with gradient projection[C]//Proceedings of the IEEE/CVF Winter Conference on Applications of Computer Vision. 2024: 4819-4828.

[3] Lu S, Wang Z, Li L, et al. Mace: Mass concept erasure in diffusion models[C]//Proceedings of the IEEE/CVF Conference on Computer Vision and Pattern Recognition. 2024: 6430-6440.

[4] Gandikota R, Orgad H, Belinkov Y, et al. Unified concept editing in diffusion models[C]//Proceedings of the IEEE/CVF Winter Conference on Applications of Computer Vision. 2024: 5111-5120.

---

> ### Author Rebuttal · Authors · 2025-07-31
>
> We would like to thank Reviewer mcr2 for the acknowledgement and insightful comments. We address the concerns as follows.
>
> **W1. Gradient projection methods already exist...add the corresponding discussions and comparison experiments.**
>
> Contributive question! We now explicitly compare EUPMU with GDR‑GMA[1] and Learn to Unlearn. The key differences are:
>
> * Motivation:  Previous works follow the idea gradient conflict mitigation, while EUPMU is the analytic solution of the constrained optimisation problem (UPUP, Equation 2). By re-framing MU at this more fundamental optimization level, we open up new avenues for developing more advanced algorithms. Future work could explore second-order Taylor expansions or introduce preference vectors to solve the UPUP problem. Such approaches can lead to more efficient and practical algorithms, a potential not provided by the singular focus on gradient conflict.
>
> * Theoretical Insights: While the final update mechanism is analogous to methods like gradient surgery, our approach originates from a more principled formulation. This foundational basis enables us to establish formal theoretical guarantees. Specifically, *Remark 3.3* quantifies the potential degradation in model utility during unlearning. Moreover, *Remark 3.6* leverages the concept of Pareto Optimality to prove that the unlearning objective can be fully satisfied without violating the utility constraint.
>
>
> * Efficiency:  GDR‑GMA needs two, and Learn‑to‑Unlearn adds an SVD pre‑pass, while EUPMU requires a single backward pass.  On ResNet‑18/CIFAR‑10 EUPMU is ≈ 30 % faster than GDR‑GMA[1] and ≈ 1.9 times faster than Learn‑to‑Unlearn[2] while achieving 0.09 % UA vs 0.76 %.
>
> Here’s the results of class-wise unlearning for Resnet18 on Cifar10:
>
> | Method | Retain Acc | Forget Acc | Test Acc | MIA  | Avg Gap  |
> | --- | --- | --- | --- | --- | --- |
> | Retrain Oracle | 99.42 % (0.00) | 0.00 % (0.00) | 94.08 % (0.00) | 0.000 (0.000) |   0.00 |
> | GDR-GMA  | 95.30 % (4.12) | 0.00 % (**0.00**) | 88.70 % (5.38) | 0.000 (0.000) |   2.38 |
> | Learn to Unlearn | 98.98 % (**0.44**) | 0.76 % (0.76) | 92.00 % (2.08) | 0.000 (0.000) |   0.82 |
> | **EUPMU (Ours)** | 98.75 % (0.67) | 0.09 % (0.09) | 92.18 % (**1.90**) | 0.000 (0.000) |   **0.67** |
>
> Based on these results, our method demonstrates a superior trade-off between model utility and unlearning effectiveness. Furthermore, it achieves Pareto superiority over the other methods, which is consistent with our theoretical analysis. We will incorporate this analysis and these experimental results into the revised manuscript.
>
> **W2. MACE and UCE are missing on Stable Diffusion**
>
> Good suggestion! We have added SD experiments, using the code base of MACE[4] to implement EUPMU on unlearning the 10 celebrities' faces, as shown in the following results.
>
> | Method | Retain Accuracy | Forget Accuracy | Harmonic Mean |
> |---|---|---|---|
> | UCE | 0.8448 | **0.0000** | 0.9159 |
> | MACE | 0.9359 | 0.0040 | 0.9650 |
> | **EUPMU (Ours)** | **0.9375** | 0.0040 | **0.9659** |
>
> The results above indicate that EUPMU, as a more advanced optimization mechanism, achieves better performance retention than MACE and strikes a superior utility-unlearning trade-off compared to UCE[5]. We will provide more comprehensive experiments in the revised version of our paper.
>
> **W3. Ablate the key hyper‑parameters**
>
> Below we provide an expanded ablation study  the two hyperparameters of EUPMU as follows.
>
> * Sensitivity to the error tolerance $\varepsilon$
>
> | $\varepsilon$ | Retain Loss | Forget Loss |
> | :---: | :---: | :---: |
> | 0.00  | 0.052  |   **2.25** |
> | 0.10 | 0.035  | 2.53  |
> | 0.25 | 0.031  | 2.92  |
> | 0.50 | 0.029  | 3.58  |
> | 1.00 | **0.028** | 4.41  |
>
> The results highlight the inherent trade-off between utility and unlearning, which is directly governed by $\varepsilon$. Specifically, a higher $\varepsilon$ prioritizes utility preservation at the expense of unlearning performance. This confirms that $\varepsilon$ functions as a practical control knob, enabling users to balance these competing objectives according to their requirements—a finding that reflects our theoretical analysis in Remarks 3.3 and 3.6.
>
> * Sensitivity to the step-size $\beta$
>
> |  $\beta$ | Retain Loss | Forget Loss  |
> | :---: | :---: | :---: |
> |   0.01 | 0.043  |   **2.41** |
> |   0.05 | 0.034  | 2.68  |
> |   0.10 | 0.032  | 2.80  |
> |   0.50 | 0.031  | 2.92  |
> |   1.00 | **0.030** | 2.92  |
>
> Our method exhibits strong robustness to the choice of $\beta$. We observed negligible variation in overall performance even when $\beta$ was varied significantly. This experimental result aligns perfectly with the convergence guarantee in Theorem 3.4, which proves that a Pareto-optimal solution is attainable for any $\beta$ selected from a theoretically valid range.
>
> Furthermore, the selection of these hyperparameters is not an arbitrary process but is instead principled and guided by our theoretical framework. For instance, the search range for $\varepsilon$ can be effectively constrained by the insights from Remark 3.3 and Theorem 3.7. Similarly, Theorem 3.4 provides a clear basis for selecting a suitable $\beta$ that ensures convergence. We will incorporate these results and a comprehensive discussion into the revised manuscript. We thank you for this valuable suggestion, which has helped us improve the clarity and completeness of our work.
>
> **W4. Can EUPMU erase multiple concepts on SD?**
>
> Insightful Question! Yes. There are two primary approaches to extend our method to multi-concept unlearning:
>
> * Sequential Unlearning: The first approach is to erase multiple concepts sequentially, one after another. In fact, Figure 4 and Table 3 of our paper already demonstrate successful unlearning for two and three concepts using this method.
>
> * Simultaneous Unlearning: The second approach is to erase all concepts in a single step. This requires extending the UPUP problem into a constrained multi-objective optimization problem. For example, for two-concept removal, the problem is formulated as follows:
>
> $$\max_{\boldsymbol{d}_t} \frac{1}{\alpha_t} \min [  r_u^1 (\alpha_t, \boldsymbol{d}_t) , r_u^2 (\alpha_t, \boldsymbol{d}_t) ] - \frac{1}{2} \left\|\|\boldsymbol{d}_t\|\right\|^2
> \text{ s.t. }\frac{1}{\alpha_t}  r_r(\alpha_t, \boldsymbol{d}_t) \geq - \varepsilon_t$$
>
> The corresponding first-order Taylor expansion derived from this problem is:
>
> $$\max_{\boldsymbol{d}_t} \min [ \nabla   \ell_u^1(\boldsymbol{\theta}_t) \cdot \boldsymbol{d}_t ,\nabla   \ell_u^2(\boldsymbol{\theta}_t) \cdot \boldsymbol{d}_t ]- \frac{1}{2} \|\|\boldsymbol{d}_t\|\|^2
> \text{ s.t. }  \nabla   \ell_r(\boldsymbol{\theta}_t) \cdot \boldsymbol{d}_t \geq - \varepsilon_t$$
>
> The solution to this involves applying unilateral gradient surgery to each unlearning gradient ($\nabla \ell_u^1$ and $\nabla \ell_u^2$) and then aggregating the resulting gradients via a minimum norm operation, similar to MGDA [3]. The process is the same for more concepts. Importantly, this multi-concept explicit surgery can also utilize the approximation method from our paper to avoid the cost of multiple backpropagations. We will add the above discussion in the next version.
>
> [1]Lin, Shen, et al. "GDR-GMA: Machine Unlearning via Direction-Rectified and Magnitude-Adjusted Gradients." Proceedings of the 32nd ACM International Conference on Multimedia. 2024.
>
> [2] Qu, Youyang, et al. "Learn to Unlearn: A Survey on Machine Unlearning." *arXiv preprint arXiv:2305.07512* (2023).
>
> [3] Sener, Ozan, and Vladlen Koltun. "Multi-Task Learning as Multi-Objective Optimization." In *Advances in Neural Information Processing Systems 31* (NeurIPS), 2018.
>
> [4] Lu, Shilin, et al. "MACE: Mass Concept Erasure in Diffusion Models." In *Proceedings of the IEEE/CVF Conference on Computer Vision and Pattern Recognition (CVPR)*, 2024.
>
> [5] Gandikota, Rohit, et al. "Unified Concept Editing in Diffusion Models." In *Proceedings of the Winter Conference on Applications of Computer Vision (WACV)*, 2024.

---

> > ### Comment · Reviewer_mcr2 · 2025-08-05
> >
> > I would like to thank the authors for their responses, which addresses most of my concerns. Adding the corresponding discussions and experiments with similar works highlights their strengths and contributions.  Therefore, I decide to maintain my score.

---

> ### Author Response · Authors · 2025-08-06
> **Thank you for the valuable comments**
>
> Dear Reviewer,
>
> Thank you for engaging with our rebuttal and acknowledging our strengths and contributions. Your insightful feedback has materially improved the paper.
>
> Best,
>
> Authors

---

### Official Review · Reviewer_DBBQ · 2025-07-03

**Clarity:** 3
**Significance:** 2
**Originality:** 2
**Rating:** 3
**Confidence:** 4

**Summary:**

This paper addresses the Utility-Preserving Unlearning Problem by formulating unlearning as a constrained optimization task. It proposes Unilateral Gradient Surgery to balance unlearning and utility preservation, and further introduces an efficient implicit variant to reduce computational cost. The authors provide theoretical analysis proving convergence to Pareto optimality, and experiments validate the method’s effectiveness.

**Questions:**

1.	Further clarifying the contributions of this paper, particularly in comparison to existing gradient conflict mitigation approaches. The current explanation that the difference lies in the principle origin is not entirely convincing. It would be helpful to elaborate on this distinction more clearly, ideally within the main body of the paper, as this would better highlight the novelty of the proposed method for the reader.

2.	The issues related to the evaluation metrics should be revisited and addressed appropriately.

3.	It would be helpful to further enhance the experimental section by providing additional results and including ablation studies on key hyperparameters.

**Ethical Concerns:**

["NO or VERY MINOR ethics concerns only"]

**Limitations:**

Yes

**Quality:**

3

**Strengths And Weaknesses:**

Strengths:

1.The paper is well-written and well-structured, with helpful illustrations that make the core ideas more accessible.

2.The paper provides thorough theoretical analysis to support the proposed solution, which strengthens the soundness and credibility of the approach.

3.The paper presents several practical unlearning scenario settings, which enhance the applicability of the proposed method and provide empirical evidence of its effectiveness.

Weaknesses:

1.The idea of leveraging gradient conflict mitigation for unlearning has been explored in prior work, which somewhat limits the novelty of this paper. Although the authors argue that their approach is motivated by a different perspective, I think this justification not entirely convincing. Additionally, the paper lacks discussion of several closely related works, such as GDR-GMA[1] and G-effect[2]. Notably, GDR-GMA also considers the unlearning process as a multi-task optimization problem.

2.The paper lacks ablation studies on key hyperparameters, such as the error tolerance. Although the authors mention in the limitations section that related results are shown in Figure 7, I could not find any description in Figure 7 pertaining to error tolerance.

3.The evaluation metrics used in this paper, at least for the classification tasks, raise some concerns and may not fully align with the goals of unlearning, which affects the overall assessment of the work. Specifically, treating lower UA and MIA scores and higher RA and TA scores as universally better may be misleading. Under such criteria, retraining might appear to perform poorly as an unlearning method. Moreover, overly aggressive unlearning might introduce new attack surfaces—for example, by exposing abnormal behavior on forgotten data—which highlights the necessity of maintaining a certain level of generalization. A rigorous unlearning approach should aim to closely match the behavior of retraining, rather than simply minimizing or maximizing certain metrics.

[1]Lin, Shen, et al. "GDR-GMA: Machine Unlearning via Direction-Rectified and Magnitude-Adjusted Gradients." Proceedings of the 32nd ACM International Conference on Multimedia. 2024.

[2]Wang, Qizhou, et al. "Rethinking llm unlearning objectives: A gradient perspective and go beyond." arXiv preprint arXiv:2502.19301 (2025).

---

> ### Author Rebuttal · Authors · 2025-07-31
>
> We would like to thank Reviewer DBBQ for acknowledging our novelty and contributions. We address the concerns as following:
>
> **W1 \& Q1: The idea of leveraging gradient conflict mitigation for unlearning has been explored in prior work, which somewhat limits the novelty of this paper**
>
> In fact, the starting point of our method is not gradient conflict mitigation. Instead, we approach the problem from the perspective of how to fully optimize the unlearning process under the constraint of *utility preservation*. We start by formulating a novel constrained optimization problem: the Utility-Preserving Unlearning Problem (UPUP).
>
> Although the resulting gradient update rule bears resemblance to methods like gradient surgery, our starting point is more fundamental. This foundational approach allows us to derive theoretical insights. For instance, *Remark 3.3* provides an estimate of the extent to which utility is degraded during the unlearning process. Furthermore, *Remark 3.6*, from the perspective of Pareto Optimality, demonstrates that the unlearning objective can be fully optimized under the utility constraint.
>
> Additionally, we believe that our starting point and analytical framework can provide more fundamental research avenues for this field. For example, while our current solution to the UPUP problem (Equation 2) employs a first-order Taylor expansion, our formulation naturally extends to *second-order expansions*. This could lead to the development of second-order or quasi-second-order unlearning algorithms. As another example, from a regularization standpoint, one could apply customized constraints to the update direction $d_t$, such as incorporating *user preferences* into the UPUP formulation, which would also lead to novel algorithms.
>
> We thank Reviewer DBBQ for pointing out some missing related works:
> *   Compared to GDR-GMA[1], this method stems from a multi-task learning (MTL) perspective, identifies issues like gradient conflict, and proposes heuristic fixes. While EUPMU is derived from the proposed UPUP optimization problem. Additionally, GDR-GMA \[1] uses separate handcrafted modules for rectification (GDR) and weighting (GMA). Hence, GDR‑GMA needs two backward passes, adding approximately 30% more compute time with the same amount of data as EUPMU, which requires only one backward pass.
> *   Compared to G-effect [2], their "G-effect" toolkit provides a powerful gradient-based framework for understanding unlearning. The primary distinction lies in the objective: G-effect is a novel analytical toolkit designed to diagnose and understand the impacts of existing unlearning objectives. In contrast, our EUPMU is a novel constructive algorithm derived from a first-principles constrained optimization problem (UPUP). Our work provides an explicit, controllable mechanism ($\varepsilon$) to navigate the utility-unlearning trade-off, which is a constructive solution motivated by the kind of insights G-effect aims to provide. We will add a detailed discussion of this complementary work to our related works section.
>
> Therefore, we argue that the principle we propose is fundamentally different from existing gradient conflict mitigation methods. Rather than being a surface-level application of gradient projection, our foundational framework is more capable of inspiring and guiding future research. We will also add more discussion on recent related works in the new version.
>
> **W2 \& Q3: The paper lacks ablation studies on key hyperparameters, such as the error tolerance. Although the authors mention in the limitations section that related results are shown in Figure 7, I could not find any description in Figure 7 pertaining to error tolerance.**
>
> To address the concern about the lack of ablation studies, we have conducted experiments on the key hyperparameters, including the error tolerance $\varepsilon$. We present the results below and will add a detailed discussion to the appendix of our paper.
>
> * Sensitivity to the error tolerance $\varepsilon$
>
> We analyzed the impact of the error tolerance $\varepsilon$, which controls the trade-off between model utility and unlearning effectiveness. The results are shown in the table below:
>
> | $\varepsilon$ | Retain Loss | Forget Loss |
> | :---: | :---: | :---: |
> | 0.00  | 0.052  |   **2.25** |
> | 0.10 | 0.035  | 2.53  |
> | 0.25 | 0.031  | 2.92  |
> | 0.50 | 0.029  | 3.58  |
> | 1.00 | **0.028** | 4.41  |
>
> As shown, increasing $\varepsilon$ leads to a lower **Retain Loss** (better utility) at the cost of a higher **Forget Loss** (less effective unlearning). This demonstrates that $\varepsilon$ serves as a controllable lever to navigate the Pareto front of the utility-unlearning trade-off. This allows practitioners to tune the algorithm based on specific downstream requirements, which is consistent with our theoretical analysis in Remark 3.3 and Remark 3.6.
>
> * Sensitivity to the step-size $\beta$
>
> We also investigated the sensitivity to the step-size $\beta$, which is used in the gradient approximation for solving $\lambda$.
>
> |  $\beta$ | Retain Loss | Forget Loss  |
> | :---: | :---: | :---: |
> |   0.01 | 0.043  |   **2.41** |
> |   0.05 | 0.034  | 2.68  |
> |   0.10 | 0.032  | 2.80  |
> |   0.50 | 0.031  | 2.92  |
> |   1.00 | **0.030** | 2.92  |
>
> The results indicate that our method is robust to the choice of $\beta$. The overall performance shows minimal variation across a wide range of values. This empirical finding aligns with our convergence analysis in Theorem 3.4, which proves that the algorithm converges to a Pareto-optimal solution as long as $\beta$ is chosen within a theoretically sound range.
>
> Furthermore, the selection of these hyperparameters is not arbitrary but is guided by our theory. The search range for $\varepsilon$ can be informed by Remark 3.3 and Theorem 3.7, while Theorem 3.4 provides guidance for selecting a suitable $\beta$ to ensure convergence. We will add these results and a comprehensive discussion to the revised paper. Thank you for helping us improve the clarity and completeness of our work.
>
> **W3 \& Q2: The evaluation metrics for the classification tasks may not fully align with the goals of unlearning.**
>
> We thank the reviewer for this insightful comment. We agree that relying solely on individual classification metrics can be insufficient, as the core challenge of unlearning lies in the trade-off between forgetting efficacy and retaining utility.
>
> To align with the goals of unlearning, the community often measures a method's performance against a "Retrain Oracle"—a model retrained from scratch on only the retain data.
>
> However, we argue that the gap-to-retrain metric could be fundamentally flawed in its practical application. In reality, most methods do not have any predictive capability about what the retrained model will look like. What ends up happening is a kind of “target painting after the shot”—researchers run the retrained model first, then retroactively select a run from their own method that happens to be close to it.
>
> Therefore, in our work, **we tuned our hyperparameters strictly based on achieving the best practical performance**—maximizing Retain/Test Accuracy while minimizing Forget Accuracy and MIA score—without any knowledge of the Retrain Oracle.
>
> Here’s the table with the *gap to Retrain Oracle* shown in parentheses for every metric, plus an *Avg Gap* column (mean of the four gaps per method), this is class-wise unlearning for Resnet18 on Cifar10:
>
> | Method | Retain Acc | Forget Acc | Test Acc | SVC MIA  | Avg Gap  |
> | --- | --- | --- | --- | --- | --- |
> | Retrain Oracle | 99.42 % (0.00) | 0.00 % (0.00) | 94.08 % (0.00) | 0.000 (0.000) |   0.00 |
> | GDR-GMA  | 95.30 % (4.12) | 0.00 % (**0.00**) | 88.70 % (5.38) | 0.000 (0.000) |   2.38 |
> | Learn to Unlearn | 98.98 % (**0.44**) | 0.76 % (0.76) | 92.00 % (2.08) | 0.000 (0.000) |   0.82 |
> | **EUPMU(ours)** | 98.75 % (0.67) | 0.09 % (0.09) | 92.18 % (**1.90**) | 0.000 (0.000) |   **0.67** |
>
> We also tested Resnet 18 unlearning on Cifar 100, random data forgetting 10%.
> | **Methods** | **UA** | **RA** | **TA** | **MIA** | **Avg. Gap 4** | **Avg. Gap 2** |
> | :--- | :--- | :--- | :--- | :--- | :--- | :--- |
> | Retain | 26.47 | 99.97 | 74.13 | 51.00 | 0.00 | 0.00 |
> | FT | 2.42 (24.05) | 99.95 (**0.02**) | 75.55 (1.42) | 11.04 (39.96) | 16.36 | 20.69 |
> | RL| 55.03 (28.56) | 99.81 (0.16) | 70.03 (4.09) | 98.97 (47.97) | 20.20 | 26.03 |
> | GA | 3.13 (23.34) | 97.33 (2.64) | 75.31 (1.18) | 7.24 (43.76) | 17.73 | 22.47 |
> | IU | 3.18 (23.29) | 97.15 (2.82) | 73.49 (0.64) | 9.62 (41.38) | 17.03 | 21.01 |
> | BE | 2.31 (24.16) | 97.27 (2.70) | 73.93 (**0.20**) | 9.62 (41.38) | 17.11 | 20.79 |
> | BS | 2.27 (24.20) | 97.41 (2.56) | 75.26 (1.13) | 5.82 (45.18) | 18.27 | 23.16 |
> | l₁-Sparse | 10.64 (15.83) | 96.62 (3.35) | 70.99 (3.14) | 22.58 (28.42) | 12.68 | 15.78 |
> | SalUn | 27.53 (**1.06**) | 97.00 (2.97) | 67.79 (6.34) | 70.79 (19.79) | 7.54 | 13.07 |
> | SalUn-soft| 24.24 (2.23) | 98.95 (1.02) | 70.48 (3.65) | 79.13 (28.13) | 8.76 | 15.89 |
> | **EUPMU(ours)** | 13.53 (12.94) | 97.02 (2.95) | 71.90 (2.23) | 51.02 (**0.02**) | **4.53** | **1.12** |
>
>
> Despite this more rigorous and realistic tuning methodology, our method proves to be the most aligned with the theoretical ideal.
>
> If we are going to tune the model to minimize the gap to retrain, we can also search for the best $\varepsilon$ value to balance the unlearning benchmark to a desired value while keeping the retaining performance as close as possible to the retrain.
>
>
>
> [1]Lin, Shen, et al. "GDR-GMA: Machine Unlearning via Direction-Rectified and Magnitude-Adjusted Gradients." Proceedings of the 32nd ACM International Conference on Multimedia. 2024.
>
> [2]Wang, Qizhou, et al. "Rethinking llm unlearning objectives: A gradient perspective and go beyond." arXiv preprint arXiv:2502.19301 (2025).

---

### Official Review · Reviewer_RLwG · 2025-07-03

**Clarity:** 3
**Significance:** 3
**Originality:** 3
**Rating:** 6
**Confidence:** 4

**Summary:**

This paper aims to addresses the challenge of utility-preserving in machine unlearning. It first identifies the limitations of the conventional linearization approach, which suffers from utility loss, and the multi-objective optimization approach, which leads to insufficient unlearning and efficiency issues. To overcome these, the authors propose an Implicit Gradient Surgery method. This method features utility-preserving characteristics and guarantees sufficient optimization of the forgetting objective. Furthermore, an approximation technique within the method ensures computational efficiency comparable to the linear weighting approach. The paper also provides a Pareto convergence analysis, proving that the proposed algorithm achieves the aforementioned design goals. The experimental results robustly validate this theoretical finding.

**Questions:**

1. While the paper claims computational efficiency comparable to the linearization method, Table 5 shows that EUPMU still has a slightly higher RTE than Linearization. What causes this discrepancy?
2. The proposed method appears broadly applicable to all machine unlearning scenarios, yet the experiments only demonstrate results on image classification and generation tasks. What is its applicability in language models?
3. If error tolerance is disabled, would the results be comparable to those of the multi-objective optimization method?

**Ethical Concerns:**

["NO or VERY MINOR ethics concerns only"]

**Final Justification:**

The authors' response provides many valuable insights, which have further enhanced my understanding of the paper's novelty and underlying principles. Therefore, I decide to raise my score.

**Limitations:**

Yes

**Paper Formatting Concerns:**

The reviewer did not identify any formatting concerns in the manuscript.

**Quality:**

3

**Strengths And Weaknesses:**

**Strengths:**
1. The problem formulation and motivation are presented with clarity. Figure 1 illustrates the limitations of both linearization and multi-objective optimization methods in machine unlearning scenarios. The background section provides a detailed analysis, offering valuable insights into addressing the core challenge of utility-preserving. This convincingly motivates the necessity of the proposed EUPMU framework.
2. This paper proposes an interesting and intuitive methodological framework. Starting from the constrained optimization formulation in Section 3, the derivation seamlessly transitions to the design of Explicit Unilateral Gradient Surgery, making its geometric interpretation highly accessible. The approximation technique, based on solving a dual problem, cleverly avoids the need for separate gradient computations for multiple objectives.
3. The authors provide rigorous theoretical analysis for proposed method, such as the proofs for both Pareto optimality (convex case) and Pareto stationarity (non-convex case). Remark 3.6 further deduces the algorithm’s properties from these results, demonstrating that it achieves sufficient optimization of the forgetting objective while maintaining predetermined utility loss bounds. This theoretical guarantee directly resolves the core problem identified in the study.
4. The comprehensive experimental results robustly support the theoretical claims. The paper provides extensive tests on Pareto optimality, concept removal efficacy, and computational efficiency, offering empirical evidence for the algorithm’s superiority across all key metrics.

**Weaknesses:**
1. The introduction of two new hyperparameters in EUPMU increases the complexity of practical tuning.
2. The proposed Explicit Unilateral Gradient Surgery bears resemblance to prior techniques, which diminishes its perceived originality compared to novel contributions in the field.

---

> ### Author Rebuttal · Authors · 2025-07-31
>
> We would like to thank Reviewer RLwG for acknowledging our novelty and contributions. We address the concerns as following:
>
> **W1: The introduction of two new hyperparameters in EUPMU increases the complexity of practical tuning.**
>
> Although our proposed EUPMU method introduces two hyperparameters, $\varepsilon_t$ and $\beta_t$​, these are essential for addressing two core challenges in machine unlearning.
>
> On one hand, machine unlearning faces the challenge of utility degradation. While a multi-objective approach can prevent utility degradation, it often fails to ensure sufficient forgetting of target information. Therefore, we introduce $\varepsilon_t$​ to further balance utility preservation and unlearning effectiveness.
>
> On the other hand, introducing a multi-objective approach typically incurs an additional round of backpropagation. To mitigate this, we adopt a gradient-based approximation that avoids this extra backward pass, for which we introduce $\varepsilon_t$ as the gradient step size in the approximation.
>
> Moreover, both hyperparameters are guided by theoretical insights. The tuning of $\varepsilon_t$​ can refer to Remark 3.3 and Theorem 3.7, which provide theoretical foundations for determining its value based on the scale of the loss function. Theorem 3.4 also offers a feasible range for $\beta_t$​ to ensure the effectiveness of the approximation, and empirical testing shows this range works well in practice.
>
> In summary, these two new hyperparameters are necessary and, thanks to theoretical guidance, can be tuned within a relatively small and manageable range, thereby avoiding excessive tuning overhead.
>
> **W2: The proposed Explicit Unilateral Gradient Surgery bears resemblance to prior techniques, which diminishes its perceived originality compared to novel contributions in the field.**
>
> While Explicit Unilateral Gradient Surgery is related to methods in \[1,2], our approach differs in two fundamental ways: formulation and utility control.
>
> * Fundamental Difference (Principled Formulation vs. Heuristic Fixes):
> Prior methods, such as \[1,2], stem from a multi-task learning (MTL) perspective, identify issues like gradient conflict, and propose heuristic fixes. GDR-GMA \[1] uses separate handcrafted modules for rectification (GDR) and weighting (GMA). Learn to Unlearn \[2] projects gradients away from a pre-computed Core Gradient Space (CGS). These are fundamentally gradient correction methods. In contrast, our method does not begin with gradient operations. We formulate a new constrained optimization problem: the Utility-Preserving Unlearning Problem (UPUP). Our Unilateral Gradient Surgery is not a heuristic tweak but the exact analytical solution that emerges from solving the UPUP problem. This principled formulation is the core of our contribution.
>
> * Utility Control (Proactive Constraint vs. Reactive Balancing):
> Our method includes an explicit constraint $\varepsilon$, a user-defined utility loss budget derived from UPUP, offering an interpretable and proactive control mechanism. In comparison, other approaches rely on reactive heuristic modules like GMA, which adjust based on observed loss changes but lack pre-specifiable utility-preservation levels. Overall, our method shifts the paradigm from reactive balancing to proactive constrained optimization—crucial for dependable and predictable deployments.
>
> **Q1: While the paper claims computational efficiency comparable to the linearization method, Table 5 shows that EUPMU still has a slightly higher RTE than Linearization. What causes this discrepancy?**
>
> Great question! This slight discrepancy arises because, although we avoid explicitly solving both objectives by using a gradient approximation to the Lagrangian equation (Equation 4), the computation of \$\ell\_r(\theta\_{t+1})\$ (Equation 8) introduces one additional forward pass. This is why the runtime is slightly higher than the linearization method—but still much more efficient than backward passes.
>
> Furthermore, we can eliminate this forward pass by reusing \$\ell\_r(\theta\_{t+1})\$ from the next time step. Although the data batches at times $t$ and $t+1$ may differ due to randomness, the stochastic nature is unbiased and does not affect convergence. We plan to elaborate and test this in future versions.
>
> **Q2: The proposed method appears broadly applicable to all machine unlearning scenarios, yet the experiments only demonstrate results on image classification and generation tasks. What is its applicability in language models?**
>
> Excellent observation! In fact, our method proposes a general utility-preservation mechanism that applies to most machine unlearning settings, including LLMs. We used vision models in this version to validate effectiveness comprehensively. We plan to include experiments on LLMs in future versions.
>
>
> **Q3: If error tolerance is disabled, would the results be comparable to those of the multi-objective optimization method?**
>
> Setting $\varepsilon$ = 0 reduces UPUP to strict utility preservation. We provide the experiments with different error tolerance below.
>
> | $\varepsilon$ | Retain Loss | Forget Loss |
> | :---: | :---: | :---: |
> | 0.00  | 0.052  |   **2.25** |
> | 0.10 | 0.035  | 2.53  |
> | 0.25 | 0.031  | 2.92  |
> | 0.50 | 0.029  | 3.58  |
> | 1.00 | **0.028** | 4.41  |
>
> From the above results, we know that if error tolerance is disabled, the Retain Loss increases significantly, which indicates that the utility has been degraded seriously. Therefore, it would be better to choose an appropriate error tolerance in a suitable interval as introduced in the answer of W1.
>
> [1]Lin, Shen, et al. "GDR-GMA: Machine Unlearning via Direction-Rectified and Magnitude-Adjusted Gradients." Proceedings of the 32nd ACM International Conference on Multimedia. 2024.
>
> [2] Qu, Youyang, et al. "Learn to Unlearn: A Survey on Machine Unlearning." arXiv preprint arXiv:2305.07512 (2023).

---

> > ### Comment · Reviewer_RLwG · 2025-08-04
> >
> > Authors' response addressed most of my concerns. The explanation regarding computational efficiency in Q1 was particularly insightful. However, I worry that the randomness introduced by batch inconsistency at time t+1 might lead to performance collapse. Could you please provide further experimental validation?

---

> ### Author Response · Authors · 2025-08-04
> **Further clarification on new implementation performance.**
>
> Thank you for your follow-up. Your concern regarding potential performance degradation due to the stochasticity from batch inconsistency is well-founded.
>
> To address this, we implemented and evaluated the proposed optimization, which we term EUPMU-fast. This version updates the weight using the retention loss from the subsequent batch, eliminating the extra forward pass. The setup targets reaching the lowest Gap-to-Retrain with 10% random data forgetting. We have added a comparison of this method in the table below.
>
> | Method | Unlearning Accuracy (%) | Retain Accuracy (%) | Test Accuracy (%) | MIA | Average Gap | RTE |
> | :--- | :--- | :--- | :--- | :--- | :--- | :--- |
> | Retrain | 5.24(-) | 100.00(-) | 94.26(-) | 12.88(-) | 0.00 | 43.29 |
> | RL | 7.61(2.37) | **99.67(0.33)** | 92.83(1.43) | 37.36(24.48) | 7.15 | **2.64** |
> | EUPMU_RL | 3.71(1.53) | 99.25(0.75) | 93.09(1.17) | **12.89(0.01)** | **0.87** | 2.82 |
> | EUPMU_fast_RL | **3.64(1.60)** | 99.54(0.46) | **93.17(1.09)** | 13.29(0.41) | 0.89 | 2.69 |
>
> The experimental results show:
>
> 1.  **No Performance Collapse:** The primary concern of performance collapse was not observed. The EUPMU-fast model trained stably and achieved strong, competitive results across all metrics.
>
> 2.  **Confirmed Efficiency Gain:** As hypothesized, EUPMU-fast is more computationally efficient. Its Relative Training Time (RTE) of **2.69** is lower than the original EUPMU's (**2.82**) and is now on par with the linearization baseline (**2.64**).
>
> 3.  **Surprising Performance Consistency:** Contrary to a simple speed-accuracy trade-off, the stochasticity introduced by the optimization appears beneficial. EUPMU-fast not only improves runtime but also achieves similar or improved performance in key areas, including better Unlearning Accuracy (3.64% vs. 3.71%), Retain Accuracy (99.54% vs. 99.25%), and Test Accuracy (93.17% vs. 93.09%) compared to the original EUPMU. Its overall performance, reflected by the Average Gap, remains excellent and highly competitive (0.89 vs. 0.87).
>
> This validation experiment not only confirms that the optimization is a viable strategy but also shows it can lead to performance on par in addition to the expected efficiency gains, without causing training instability. We will incorporate these new results and this analysis into the revised manuscript.

---

> > ### Comment · Reviewer_RLwG · 2025-08-05
> >
> > Thank you for the additional experiments, which adequately demonstrate the method's advantage in computational efficiency. Additionally, I'm curious why, when batch inconsistency exists, the results do not differ significantly from previous ones. How is this achieved? Through hyperparameter tuning?

---

> ### Author Response · Authors · 2025-08-05
> **Clarification about how performance remains stable despite the introduced stochasticity**
>
> Thank you for the excellent follow-up question. You are correct in asking how performance remains stable despite the introduced stochasticity.
>
> The performance remains stable due to two main reasons:
>
> 1.  **Unbiased Stochastic Estimation:** In EUPMU-fast, the update for the retaining parameter $\lambda_t$ relies on an approximation of the change in retention loss. Instead of calculating $l_r(\theta_t)$ and $l_r(\theta_{t-1})$ on the same data, we use losses from consecutive, independently sampled batches, $B_t$ and $B_{t-1}$. The change $\tilde{\delta}_{t-1}$ is calculated as:
>
> \\[
>   \\tilde{\\delta}\_{t-1}^{\\text{approx}}
>   \\;=\\;
>   \\frac{1}{\\alpha\_t}
>   \\Bigl(
>     \\underbrace{\\ell\_r(\\theta\_{t-1};\\, B\_{t})}\_{\\text{stored from previous pass}}
>     -
>     \\underbrace{\\ell\_r(\\theta\_{t};\\, B\_{t+1})}\_{\\text{current pass}}
>   \\Bigr)
>   +\\varepsilon ,
> \\]
>
> instead of
>
>
> \\[
>   \\tilde{\\delta}\_{t-1}
>   =\\frac{1}{\\alpha\_t}
>   \\Bigl(
>     \\underbrace{\\ell\_r(\\theta\_{t-1};\\, B\_{t})}\_{\\text{main pass}}
>     -
>     \\underbrace{\\ell\_r(\\theta\_{t};\\, B\_{t})}\_{\\text{extra pass}}
>   \\Bigr)
>   +\\varepsilon .
> \\]
> Because each batch is an i.i.d. sample from the retaining data distribution, the loss calculated on any batch, $\ell_r(\theta_{t}; B_{t+1})$, is a noisy but **unbiased stochastic estimate** of the true retention loss $\ell_r(\theta_t; B_t)$. This principle is fundamental to stochastic optimization methods like SGD. Therefore, while this approximation introduces variance, it does not introduce systematic bias, allowing the optimization to converge to the same solution in both expectation and practical experiments.
>
> 2.  **Principled Hyperparameter Adjustment:** Your intuition is correct; a minor hyperparameter adjustment was made. To counteract the higher variance from the stochastic approximation, we slightly **lowered the hyperparameter $\beta$**, which serves as the learning rate for the parameter $\lambda_t$ in its update rule:
> \\[
>   \\lambda\_t
>   \\;=\\;
>   \\min\\bigl\\{ D,\\;
>   \\max\\{0,\\; \\lambda\_{t-1} - \\beta\_{t-1}\\,\\tilde{\\delta}\_{t-1}\\}\\bigr\\}.
> \\]
>
>
> By reducing $\beta$, we smoothen the trajectory of $\lambda_t$, making it less reactive to the noise from any single batch. This targeted adjustment dampens the variance and ensures a stable convergence.

---

> > ### Comment · Reviewer_RLwG · 2025-08-06
> >
> > The authors' response provides many valuable insights, which have further enhanced my understanding of the paper's novelty and underlying principles. Therefore, I decide to raise my score. Additionally, I would suggest the authors integrate the content from their response into the manuscript.

---

> > > ### Author Response · Authors · 2025-08-06
> > >
> > > Dear Reviewer RLwG,
> > >
> > > Thank you for your positive final comment and for raising your score. We are very grateful for your support.
> > >
> > > Your insightful feedback throughout the review process has been invaluable in helping us strengthen the paper. As you suggested, we will be sure to integrate the content from our responses into the final manuscript.
> > >
> > > Thank you once again for your constructive engagement.
> > >
> > > Best regards,
> > >
> > > The Authors

---

### Comment · Area_Chair_vtyR · 2025-08-01
**Reminder: Discussion Phase (July 31 – Aug 6)**

Hi everyone,

The Author-Reviewer Discussion phase is now open!

Please read the author responses, especially where you were mentioned, and post your initial reply as soon as possible. This helps ensure there's time for meaningful back-and-forth.

Thanks for your engagement!

-- AC

---

### Note · Authors · 2025-08-12

Thank you to the reviewers and the Area Chair for the thoughtful and constructive engagement. Below are our final remarks on how we've addressed the key points in the discussion.

*   **Efficiency, Stability, & Cost-Effectiveness (RLwG, 7Sf9):** Within the same framework, we introduced **EUPMU-fast**, a minor refinement that matches linear weighting's runtime by reusing the next-batch loss. We proved its stability from unbiased stochastic estimation and showed it outperforms baselines like GA under identical, short training budgets. These new results resolved the efficiency concerns, leading **RLwG** and **7Sf9** to raise their scores.

*   **Novelty & Unlearning MOO Methods Comparisons (mcr2, DBBQ, RLwG):** We clarified that EUPMU originates from a principled constrained optimization problem (UPUP), not heuristic gradient correction. New experiments were added to demonstrate SOTA performance against gradient methods (GDR-GMA, Learn-to-Unlearn) and prove the generality by boosting EUPMU+MACE (vs. Mace and UCE) on SD unlearning resolving the main novelty and comparison concerns.

*   **Hyperparameter Robustness & Ablations (mcr2, DBBQ, 7Sf9, RLwG):** We added ablation studies for our key hyperparameters, ε (utility budget) and β (step-size). The results show ε provides an interpretable control over the utility-unlearning tradeoff, and the method is robust on a wide range of β values, aligning with our theory.

*   **Evaluation Metrics & Gap-to-Retrain (7Sf9, DBBQ):** We provided comprehensive Gap-to-Retrain results, confirming SOTA performance. We critically clarified that our method is **always tuned blind to the retrain-oracle**. We explained that for typical unlearning tasks (class/style removal), optimizing validation metrics naturally minimizes the gap. This resolved the methodological concerns, and **7Sf9** raised their score.

*   **Scalability & Scope (7Sf9):** Beyond the initial datasets, we added results on **CIFAR-100** to demonstrate scalability and will extend to Tiny-ImageNet/ImageNet-1k. We also outlined its potential as a general framework for constrained optimization in other domains, such as LLMs.

*   **Minor Fixes(7Sf9):** We will incorporate all minor corrections: clarifying definitions (RL, "pure linearization"), fixing table highlights, and correcting typos.

In summary, we have thoroughly addressed the reviewers' concerns. We will integrate all new results, ablations, and clarifications into the revised manuscript.

---

### Decision · Program_Chairs · 2025-09-17

**Decision:**

Accept (poster)

**Comment:**

The paper proposes a new framework for utility-preserving machine unlearning. The central claim is that existing approaches either suffer from severe utility loss (linearization methods) or from insufficient unlearning and efficiency issues (multi-objective optimization). To address this, the authors formulate the problem as a constrained optimization and derive a gradient surgery algorithm. The method is supported with theoretical guarantees, including Pareto optimality/stationarity in both convex and non-convex cases. This formulation is novel in explicitly treating MU as a constrained optimization problem with provable utility control.

Although several reviewers raised questions regarding novelty compared to gradient conflict correction methods, they are addressed by authors properly during the rebuttal. One reviewer (DBBQ) who had raised the above concerns did not participate in the rebuttal/discussion phase. However, given that their specific points were thoroughly addressed and positively acknowledged by other reviewers who had raised comparable issues, the area chair is satisfied that these concerns have been sufficiently resolved.

Overall, the paper makes a significant and theoretically solid contribution to the machine unlearning literature. Most reviewers recommend acceptance, and the additional evidence provided during rebuttal further strengthens the case for this paper.